# Empirical Privacy Variance

Yuzheng Hu [* 1]   Fan Wu [* 1]   Ruicheng Xian [1]   Yuhang Liu [2]   Lydia Zakynthinou [3]   Pritish Kamath [4]
Chiyuan Zhang [4]   David Forsyth [1]

## Abstract

We propose the notion of *empirical privacy variance* and study it in the context of differentially private fine-tuning of language models. Specifically, we show that models calibrated to the same $(\varepsilon, \delta)$-DP guarantee using DP-SGD with different hyperparameter configurations can exhibit significant variations in empirical privacy, which we quantify through the lens of memorization. We investigate the generality of this phenomenon across multiple dimensions and discuss why it is surprising and relevant. Through regression analysis, we examine how individual and composite hyperparameters influence empirical privacy. The results reveal a no-free-lunch trade-off: existing practices of hyperparameter tuning in DP-SGD, which focus on optimizing utility under a fixed privacy budget, often come at the expense of empirical privacy. To address this, we propose refined heuristics for hyperparameter selection that explicitly account for empirical privacy, showing that they are both precise and practically useful. Finally, we take preliminary steps to understand empirical privacy variance. We propose two hypotheses, identify limitations in existing techniques like privacy auditing, and outline open questions for future research.

## 1. Introduction

Modern large language models (LLMs) demonstrate remarkable proficiency on a wide range of tasks, from traditional ones such as summarization, to complex problem solving that involves reasoning and coding (Stiennon et al., 2020; Wang et al., 2023; Roziere et al., 2023); these capabilities arise from large-scale pre-training on massive datasets (Dubey et al., 2024; Gemma Team et al., 2024; Liu et al., 2024). To enhance domain specialization and personalization, a common practice is to fine-tune pre-trained models on downstream tasks with user-contributed data (Guo & Yu, 2022; Li et al., 2024). However, this process introduces significant privacy concerns, as datasets often contain sensitive information of individuals or organizations, which could be memorized and potentially divulged by LLMs (Carlini et al., 2021; 2023; Lukas et al., 2023; Biderman et al., 2024; Prashanth et al., 2025).

In response, differential privacy (DP; Dwork et al., 2006), a widely-adopted standard for privacy protection, has been incorporated into various stages of the LLM training pipeline, leading to a fruitful line of research advancing the utility-privacy trade-off in LLMs (Anil et al., 2021; Yu et al., 2022; Li et al., 2022; Bu et al., 2023; Wu et al., 2024). Nevertheless, privacy in LLMs is a nuanced concept, stemming not only from the unstructured and context-dependent nature of private information in natural language (Brown et al., 2022), but also from the generative nature of these models: during real-time user-model interactions, LLMs can inadvertently regurgitate private information, and such leaks are immediately made apparent to users (Sebastian, 2023; Falcão & Canedo, 2024). This reflects a pragmatic view on privacy centering on *perceptions of model behaviors*, which we term *empirical privacy*,[1] in contrast to the theoretically-grounded definition of DP. The gap between DP's theoretical guarantees and empirical privacy concerns surrounding LLMs has significant implications: research shows that people can understand better the implications of DP than its formal definitions (Xiong et al., 2020), and failures to effectively communicate DP's promise can discourage data sharing (Cummings et al., 2021; Nanayakkara et al., 2023).

---

[1]While the term empirical privacy is usually associated with privacy attacks (Fredrikson et al., 2015; Shokri et al., 2017; Balle et al., 2022) in the literature (Cormode et al., 2013; Andrew et al., 2024), our definition is broader and more aligned with LLMs: it extends the existing notion by framing vulnerability against attacks as a model behavior and shifts from worst-case threat models to practical, user-focused metrics that reflect tangible privacy risks.

---

*Equal contribution. Work done in part while at Simons Institute. [1]Siebel School of Computing and Data Science, University of Illinois Urbana-Champaign [2] Institute of Automation, Chinese Academy of Sciences [3]Department of Electrical Engineering and Computer Science, University of California Berkeley [4]Google Research. Correspondence to: Yuzheng Hu <yh46@illinois.edu>, Fan Wu <fanw6@illinois.edu>.

*Proceedings of the 42$^{nd}$ International Conference on Machine Learning*, Vancouver, Canada. PMLR 267, 2025. Copyright 2025 by the author(s).

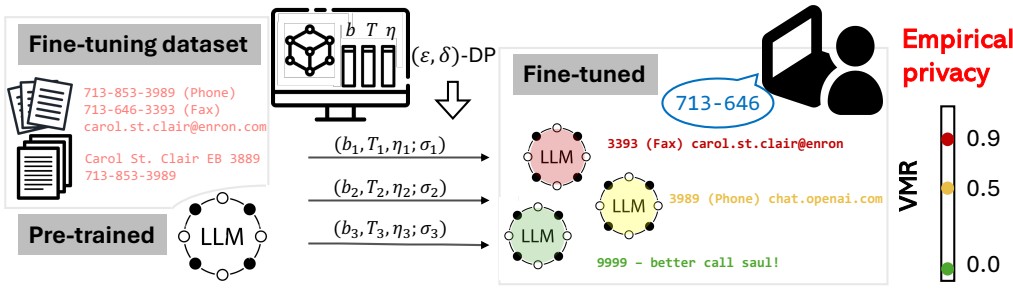

Figure 1. **Empirical privacy variance**: Starting from the *same* pre-trained model and fine-tuning on the *same* dataset (to achieve decent utility), DP-SGD with *different* hyperparameter configurations—each calibrated to the *same* $(\varepsilon, \delta)$-DP guarantee—produces models with *drastically different* privacy behaviors.

In this paper, we take an initial step toward bridging this gap by investigating the *consistency* of DP with respect to empirical privacy. Specifically, we ask: *Do LLMs calibrated to the same DP guarantee share similar levels of empirical privacy?* To explore this, we fine-tune LLMs using DP-SGD (Song et al., 2013; Bassily et al., 2014; Abadi et al., 2016) with different hyperparameter configurations, ensuring they achieve the same DP guarantee, and quantitatively assess their empirical privacy through the lens of memorization. Our results reveal a concerning inconsistency, which we refer to as *empirical privacy variance* (Fig. 1).

Our main contributions are as follows: In Section 3, we formally define our empirical privacy measures and demonstrate the phenomenon of empirical privacy variance, showing it is ubiquitous and substantial, with consistent trends across dimensions such as model, data, and privacy budget (Fig. 2). We further discuss its implications, particularly the challenges it poses for standardization. In Section 4, we analyze the influence of hyperparameters in DP-SGD on empirical privacy through regression analyses. Our findings reveal a *no-free-lunch* result: utility gains achieved from hyperparameter tuning often come at the cost of compromised empirical privacy. Based on the insights drawn from hyperparameter analyses, we propose heuristics for hyperparameter selection to improve empirical privacy and demonstrate their effectiveness. Finally, in Section 5, we explore two hypotheses underlying this phenomenon and identify key open questions to guide future research.

## 2. Preliminaries

We introduce the basics of differential privacy, DP-SGD, and memorization in language models.

**Differential privacy** (DP) is a mathematical framework that limits the information an adversary can infer about any single training example from an algorithm's output. We view the algorithm's input as a dataset consisting of *samples*; we say two datasets $D$ and $D'$ are *neighboring* if one can be obtained from the other by adding or removing a single sample.

**Definition 2.1** (Dwork et al., 2006). A randomized algorithm $\mathcal{M}$ is $(\varepsilon, \delta)$-differentially private if for all neighboring datasets $D, D'$ and for all $S \subseteq \text{Range}(\mathcal{M})$:

$$\Pr[\mathcal{M}(D) \in S] \leq e^{\varepsilon} \Pr[\mathcal{M}(D') \in S] + \delta.$$

Here, $\varepsilon$ denotes the privacy budget, with smaller values indicating stronger privacy protection, and $\delta$ is a (small) failure probability. Together, $(\varepsilon, \delta)$ are referred to as the *privacy parameters*.

**DP-SGD** (Abadi et al., 2016) is the go-to algorithm for achieving DP in deep learning and has been applied across diverse applications (De et al., 2022; Yu et al., 2022; Xu et al., 2023; Hu et al., 2024). It involves the following *training hyperparameters*: $b$ (batch size), $T$ (number of training iterations), $\eta$ (learning rate), $c$ (clipping norm). At step $t$, DP-SGD updates the model weights $w_t$ using a **privatized gradient**, obtained through per-sample gradient clipping and Gaussian noise addition:

$$\bar{g}_t := \frac{1}{|S_t|} \left( \sum_{x \in S_t} \frac{\nabla_{w_t} \ell(w_t; x)}{\max\left(1, \frac{\|\nabla_{w_t} \ell(w_t; x)\|}{c}\right)} + \mathcal{N}(0, \sigma^2 c^2 I) \right).$$

Here, $S_t$ is a mini-batch of size $b$ (see Section 7 for more discussions on the choice of samplers), $\ell$ is the loss function, and the noise multiplier $\sigma$ is computed using numerical privacy accountants (Gopi et al., 2021; Doroshenko et al., 2022) to satisfy a target $(\varepsilon, \delta)$-DP guarantee. The privatized gradient can also be used in other first-order optimizers like Adam (Kingma & Ba, 2015), leading to DP-Adam (Li et al., 2022), which we use in some experiments but refer to collectively as DP-SGD for simplicity. The full algorithms are presented in Appendix A. We additionally define $n$ (training set size) and $q := b/n$ (sampling rate) for later reference. Moving forward, we refer to a combination of training hyperparameters as a *configuration*, and an instantiation of DP-SGD with a specific configuration as a *mechanism*.

**Memorization in language models.** Memorization is

*Table 1.* Example secrets in Enron and TOFU

| Dataset | Random samples of secrets |
| --- | --- |
| Enron | "Carol St. Clair\nEB 3889\n713-853-3989"
"713-853-5620 (phone)\n713-646-3490
(fax)\nsara.shackleton@enron.com" |
| TOFU | genre ("Yevgeny Grimkov") $\longrightarrow$ "cyberpunk"
genre ("Adrianus Suharto") $\longrightarrow$ "dystopian" |

a well-documented phenomenon in LLMs (Carlini et al., 2021; Nasr et al., 2023a; 2025a). Various notions have been proposed to characterize memorization (Carlini et al., 2023; Zhang et al., 2023; Schwarzschild et al., 2024), with recent works further expanding this understanding through concepts like *approximate memorization* (Ippolito et al., 2023) and a taxonomy of memorization behaviors (Prashanth et al., 2025). In this work, we use memorization to analyze empirical privacy.

## 3. Landscape of Empirical Privacy Variance

In this section, we demonstrate empirical privacy variance across multiple dimensions and discuss its significance.

### 3.1. Experimental setups

Our experimental framework consists of two main steps: 1) fine-tuning an LLM on a dataset using DP-SGD, and 2) evaluating the empirical privacy (formally defined shortly) of the resulting model. We base our study on two sets of experiments. In the first, we fine-tune GPT-2 models (-small (S) and -large (L); Radford et al., 2019) on Enron Email (Cohen, 2004). In the second, we fine-tune Llama-2 models (-7b and -13b; Touvron et al., 2023) on TOFU (Maini et al., 2024). We ensure that the fine-tuning examples were not included in the models' pre-training data (see Appendix B.5). Below, we introduce the datasets and secrets, DP fine-tuning procedure, and empirical privacy measures.[2]

**Datasets and secrets.** The Enron Email dataset (Cohen, 2004) consists of emails by employees of the Enron Corporation. We perform a series of pre-processing steps including sample-level de-duplication (Appendix B.1), resulting in a dataset of 33k samples. We extract small pieces of sensitive information (e.g., phone numbers, see Table 1) from the dataset and define them as the *secrets*. The TOFU dataset (Maini et al., 2024) contains *synthetic* author profiles describing authors' attributes. We extract the *genre* attribute as the secret (see Table 1 and Appendix B.2) as it is relevant and easy to extract and prompt. The secret extraction procedure and the secret statistics are in Appendix B.6; we also include a discussion on the privacy unit (Appendix B.6.1).

---

[2]Our code is publicly available at https://github.com/empvv/empirical-privacy-variance.

**DP fine-tuning.** Following prior work (Wutschitz et al., 2022; Yu et al., 2022), we fine-tune LLMs with *LoRA* (Hu et al., 2022) using DP-SGD/DP-Adam (Abadi et al., 2016; Li et al., 2022), and compute a $\sigma$ that satisfies a target $(\varepsilon, \delta)$-DP guarantee using the PRV accountant (Gopi et al., 2021). We use common choices of $\varepsilon \in \{1, 2, 4, 8, 16\}$ and set $\delta = n^{-1.1}$. Finally, we evaluate the utility of the fine-tuned LLMs on a held-out test set using negative log likelihood (NLL), where lower values indicate better performance.

*Hyperparameter choices.* We perform extensive hyper-parameter tuning in the space of $(b, T, \eta)$, while fixing $c$ to a small constant, as we find that varying it within the recommended range (Li et al., 2022; De et al., 2022) has minimal impact on utility or empirical privacy. Following prior work (De et al., 2022; Ponomareva et al., 2023), we do not account for the additional privacy loss incurred by hyperparameter tuning on private data (Papernot & Steinke, 2022). For GPT-2 models on Enron, we perform a *partial* hyperparameter sweep, resulting in 23 configurations for GPT-2-S and 15 for GPT-2-L. For Llama-2 models on TOFU, we conduct a *full* grid search over $b, T, \eta$, yielding 60 configurations per setting. The difference between partial and full sweeps is due to compute constraints (see Appendix B.9). Each configuration is fine-tuned with multiple random seeds, and we retain models achieving at least 90% of the utility *gain* from the pre-trained baseline to the best-performing model. Further details on fine-tuning are deferred to Appendix B.9.

**Empirical privacy measures.** In this paper, we focus on a pragmatic view of privacy based on the perceptions of model behaviors, i.e., memorization and regurgitation of secrets. Specifically, we quantify empirical privacy through the following *memorization* scores. Let $M$ be a mapping from the input/prompt to the output/generation produced by greedy or stochastic decoding on the model.

On Enron, let $s$ denote a secret string. We consider:

- Adversarial compression ratio (ACR; Schwarzschild et al., 2024) measures how effectively a secret is stored in model weights, by optimizing for the shortest prompt eliciting it:

$$\text{ACR}(s) = \frac{|s|}{|p^*|}, \text{ where } p^* := \arg\min_p |p| \text{ s.t. } M(p) = s.$$

- Verbatim memorization ratio (VMR; adapted from Carlini et al., 2023) evaluates whether prompting with the prefix ($s_1$) of a secret leads to recovery of the remainder ($s_2$):

$$\text{VMR}(s; s_1, s_2) = \mathbb{1}[M(s_1) = s_2], \text{ where } s = s_1 \| s_2.$$

On TOFU, let $x$ be an author, $A(x)$ the author's attribute (genre), and $\mathcal{P}(x)$ a prompt aiming to elicit the secret ("What genre does $\{x\}$ write in?"). We consider:

- Attribute inference ratio (AIR; our proposed metric) measures the model's ability to recover a secret attribute in

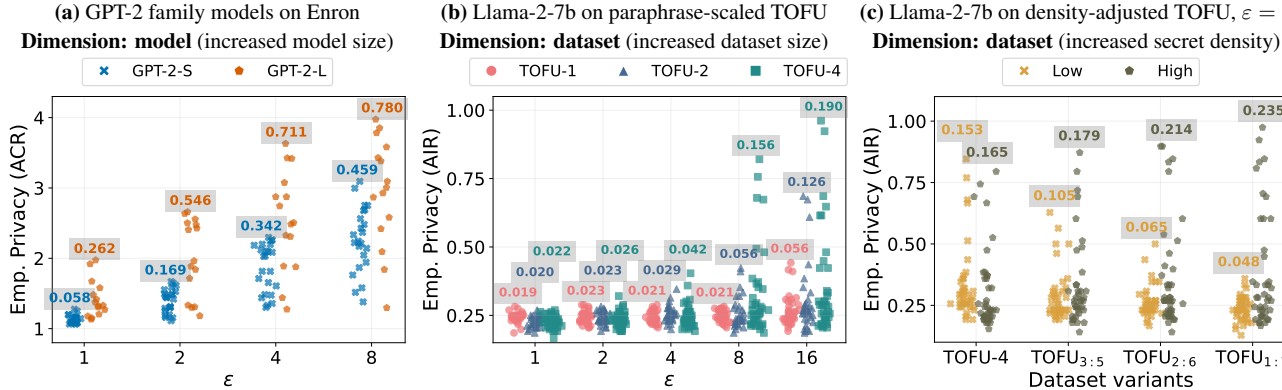

Figure 2. **Empirical privacy variance: ubiquitous, substantial, and revealing intriguing trends.** Each subfigure presents jitter plots of empirical privacy scores (ACR or AIR) obtained by models trained under a given $(\varepsilon, \delta)$-DP guarantee. Higher $y$-axis scores indicate worse empirical privacy, while the $x$-axis contrasts different groups (e.g., models of varying sizes in (a)), represented by different colors. Within each group, scattered points correspond to unique hyperparameter configurations $(b, T, \eta)$, averaged over training randomness (we show the impact of training randomness is much smaller than that of hyperparameters in Appendix C.3). Each group's *standard deviation* is labeled at the top of its cluster. The subfigures demonstrate that empirical privacy variance increases with **(a)** *model size*, **(b)** *dataset size*, **(c)** *secret density*, and **(a/b)** *privacy budget $\varepsilon$*.

response to a prompt query:

$$\text{AIR}(x) = \mathbb{1}[A(x) \text{ appears in } M(\mathcal{P}(x))].$$

We compute the average of each of these metrics (ACR, or VMR, or AIR) over a curated set of secrets, and refer to them as *empirical privacy scores*. Higher scores correspond to stronger memorization and weaker empirical privacy. Empirical privacy variance is defined as the variance of these scores in each controlled setting. Additional details about these metrics are provided in Appendix B.7.

### 3.2. Trends and generality of empirical privacy variance

Fig. 2 reveals *substantial* empirical privacy variance among high-utility models for commonly adopted $\varepsilon$ values. For instance, a Llama-2-7b trained on TOFU-4 at $\varepsilon = 8$ can either nearly fully reveal the secrets (AIR higher than 0.8) or have little knowledge of them (Fig. 2(b)). We proceed to investigate empirical privacy variance across different dimensions.

**Trends.** We analyze the trends of the variance across key dimensions in DP fine-tuning of LLMs.

*Model*: Fig. 2(a) shows that empirical privacy variance increases with model size (from 117M to 774M).

*Data*: Fig. 2(b-c) focuses on the influence of data. We generate TOFU variants with different dataset size and secret density. *Paraphrase-scaled TOFU* (TOFU-2, TOFU-4) expands the original dataset by $2\times$ and $4\times$ via paraphrasing (Appendix B.4). *Density-adjusted TOFU* applies non-uniform augmentation to two randomly partitioned groups, yielding 1:7, 2:6, 3:5 size ratios. TOFU-4 (4:4) serves as a uniform-density reference of the same size. Fig. 2(b-c) show that larger dataset or higher secret density lead to larger variance.

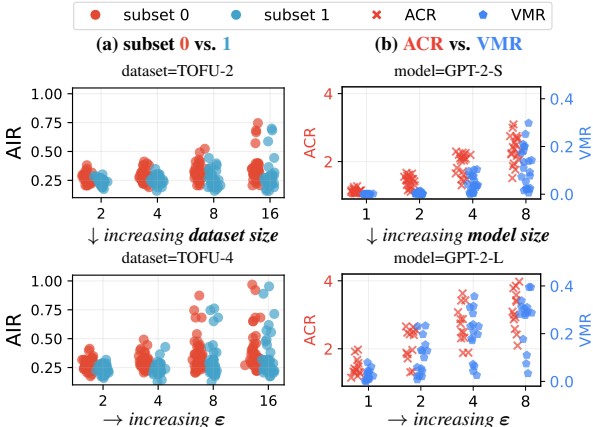

Figure 3. **Generality of empirical privacy variance.** Across **(a)** *secret subsets* (subset 0 vs. 1) and **(b)** *empirical privacy measures* (ACR vs. VMR), we observe consistent trends as in Fig. 2: empirical privacy variance increases with $\varepsilon$ ($\rightarrow$ in each subfigure), dataset size ($\downarrow$ in column (a)), and model size ($\downarrow$ in column (b)).

*Privacy budget*: Fig. 2(a-b) demonstrate a consistent trend: empirical privacy variance increases with $\varepsilon$.

*Fine-tuning paradigm*: Full fine-tuning yields higher variance than LoRA, as we show in Appendix C.1.

**Generality.** To demonstrate the generality of these trends, we examine two additional dimensions: *secret subsets* and *empirical privacy measures*. Fig. 3 shows that across these dimensions, empirical privacy variance increases with $\varepsilon$, dataset and model size, aligning with the trends observed in Fig. 2. More results are deferred to Appendix C.1.

**Intuition.** The positively contributing factors (larger models, larger paraphrased datasets, higher secret density,

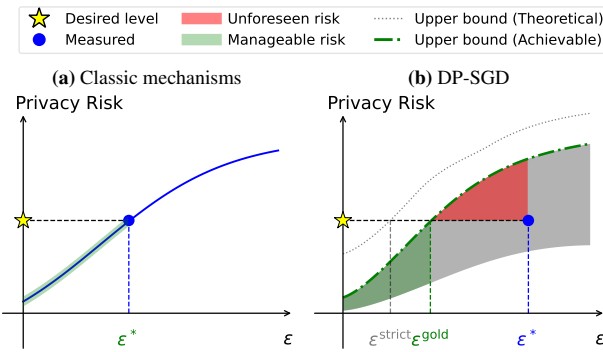

**(a)** Classic mechanisms

**(b)** DP-SGD

*Figure 4.* A *conceptual illustration* of classic mechanism vs. DP-SGD. In classic mechanisms, the monotonic relationship between privacy risk and privacy budget $\varepsilon$ allows any $\varepsilon \leq \varepsilon^*$ to be certified if $\varepsilon^*$ satisfies the desired privacy risk. In DP-SGD, however, variance introduces an *achievable region* of privacy risk, reflected by the upper and lower bound. A measured configuration meeting the privacy requirements does not safeguard the corresponding $\varepsilon^*$; identifying the truly reliable threshold, $\varepsilon^{\text{gold}}$, requires testing a wide range of configurations to account for the full spectrum of privacy risks. While a conservative *theoretical* upper bound (Yeom et al., 2018; Ma et al., 2019; Hayes et al., 2023) could aid in standardization by identifying $\varepsilon^{\text{strict}}$, such bounds are generally unavailable for empirical privacy measures like ACR.

larger $\varepsilon$) all intuitively lead to stronger memorization (Carlini et al., 2023; Ippolito et al., 2023). This intuition is empirically confirmed by our results as well, which show increasing *average* empirical privacy scores. However, a more fundamental trend we uncover is the rise in empirical privacy *variance*. We note this is a novel phenomenon and less intuitive than the increase in average scores. We defer further discussions to Appendix C.2.

### 3.3. Discussions

**Why is this surprising?** It is well-known that the interpretation of a DP guarantee heavily depends on the context: even under the same $(\varepsilon, \delta)$-DP guarantee, variations in factors like data characteristic (e.g., real-world vs. adversarially constructed, Nasr et al., 2021), model architecture (e.g., ResNet vs. CNN, Nasr et al., 2023b), and training algorithm (e.g., full vs. LoRA fine-tuning, He, 2024; Marchyok et al., 2025) can lead to different privacy implications. In contrast, we control for these factors and further restrict to models with good utility (thus avoiding trivial cases like zero updates). Despite this control, we observe substantial empirical privacy variance, highlighting the under-explored role of hyperparameters.

**Why is this relevant?** Consider classic DP mechanisms such as the Laplace and Gaussian mechanisms (Dwork et al., 2014). Their noise parameter (scale parameter $b$ for Laplace and $\sigma$ for Gaussian) inversely correlates with $\varepsilon$ and uniquely

determines privacy risk: increasing it lowers the signal-to-noise ratio, making it harder for adversaries to extract meaningful information. This establishes a one-to-one, monotonic $\varepsilon$-to-risk relationship. In contrast, the *composition* nature of DP-SGD results in a one-to-many $\varepsilon$-to-risk relationship, making $\varepsilon$ insufficient to fully capture privacy risk.

A direct consequence is that, in DP-SGD, $\varepsilon$ cannot be used for *certification*: a model calibrated to a given $\varepsilon^*$, deemed to meet privacy requirements, cannot ensure compliance for models with stricter DP guarantees ($\varepsilon \leq \varepsilon^*$). This limitation further complicates *standardization*, i.e., establishing an $\varepsilon^*$ for practitioners to follow. If a legislative body runs privacy tests (independent of $\epsilon$) and recommends $\varepsilon^*$ as a privacy standard without accounting for empirical privacy variance, there will be unforeseen risks that undermine the efficacy of such a standard (see Fig. 4 for an illustration).

## 4. How Hyperparameters Impact Empirical Privacy: Analysis and Selection Heuristics

In this section, we analyze the impact of hyperparameters through regression analyses, based on which we reveal a no-free-lunch result for empirical privacy and propose refined heuristics for hyperparameter selection. Although a linear model might not fully capture the complex relationship between empirical privacy scores and hyperparameters, we mainly use it as an exploratory tool to gain *qualitative* insights rather than definitive quantitative conclusions.

### 4.1. Dissecting effects of hyperparameters

We use `lm()` in R Statistical Software (v4.4.2) (R Core Team, 2024) to perform multivariate regression, where the target $y$ is the empirical privacy score and the covariates are the hyperparameters $b, T, \eta$. Regression is conducted in logarithmic space (log-transforming the covariates) to examine the impact of *multiplicative* changes to each hyperparameter. We focus on two settings: DP fine-tuning GPT-2-S on Enron at $\varepsilon = 4$ and Llama-2-7b on TOFU at $\varepsilon = 16$. The total number of instances for regression is 92 and 114, respectively.

**Regression on individual hyperparameters.** We regress $y$ on $(\log b, \log T, \log \eta)$. The results are presented in Table 2. The $p$-values and the coefficients indicate a statistically significant positive relationship between individual hyperparameters and empirical privacy. Additionally, the coefficient for $\log b$ is the smallest, while $\log \eta$ has the largest.

**Regression on composite hyperparameters.** We analyze the interactions between individual hyperparameters and their joint effects. Specifically, we combine $b$ and $T$ into a composite quantity called **compute** $C := b \cdot T$ (while retaining $\eta$ as a separate term due to its large coefficient), which represents the total training effort—a

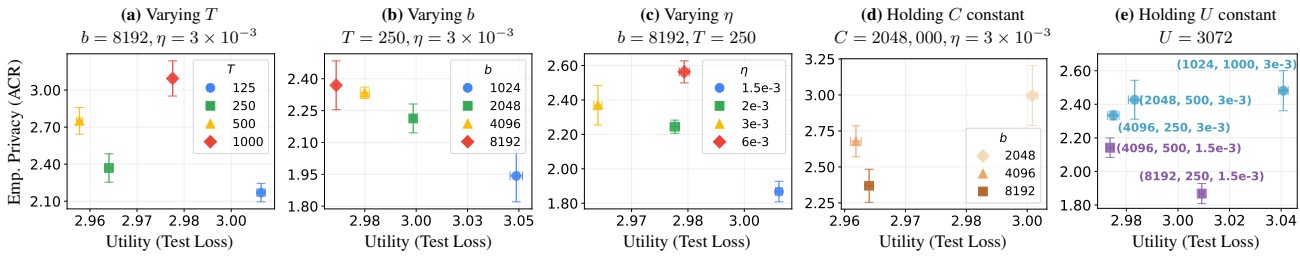

Figure 5. **Effect of individual and composite hyperparameters** (setting: GPT-2-S, Enron, ACR, $\varepsilon = 8$). We show the empirical privacy and utility of the DP fine-tuned models using different hyperparameters. **(a-c)**: Varying one hyperparameter while holding the others fixed. **(d)**: Holding compute ($C = b \cdot T$) fixed and varying $(b, T)$; **(e)**: Holding updates ($U = C \cdot \eta$) fixed and varying $(C, \eta)$.

Table 2. (a) Regression on *individual* hyperparameters

| Variable | Enron | | TOFU | |
|---|---|---|---|---|
| | Coef. | $p$-value | Coef. | $p$-value |
| Batch size ($\log b$) | 0.13*** | $1 \times 10^{-5}$ | 0.029*** | $2 \times 10^{-5}$ |
| Iterations ($\log T$) | 0.37*** | $< 2 \times 10^{-16}$ | 0.048*** | $1 \times 10^{-11}$ |
| Learning rate ($\log \eta$) | 0.51*** | $5 \times 10^{-15}$ | 0.068*** | $3 \times 10^{-12}$ |

| (b) Regression on *composite* hyperparameters | | | | |
|---|---|---|---|---|
| | Enron | | TOFU | |
| Variable | Coef. | $p$-value | Coef. | $p$-value |
| Compute ($\log C$) | 0.22*** | $2 \times 10^{-12}$ | 0.039*** | $5 \times 10^{-11}$ |
| Learning rate ($\log \eta$) | 0.53*** | $6 \times 10^{-13}$ | 0.066*** | $3 \times 10^{-11}$ |

*Notes:* ***$p < 0.001$, **$p < 0.01$, *$p < 0.05$. The response variable (empirical privacy score $y$) is ACR for Enron and AIR for TOFU, leading to different scales of the coefficients, as ACR and AIR have different ranges.

key concept in neural scaling laws (Kaplan et al., 2020; Hoffmann et al., 2022). Additionally, we define **updates** $U := C \cdot \eta$, representing the total cumulative learning signal during training. These composite hyperparameters, along with the individual hyperparameters, form a hierarchy (see Fig. 6). We regress $y$ on $(\log C, \log \eta)$. Table 2 shows that the coefficient of $\log C$ is much smaller than that of $\log \eta$.

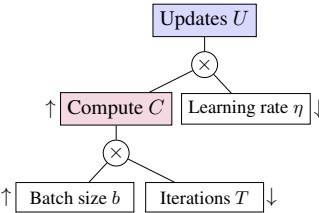

Figure 6. Hierarchy of hyperparameters. Arrows indicate the direction to improve empirical privacy when the parent node is fixed.

**Interpretations of regression results.**

- *Positive coefficients*: Increasing any individual hyperparameter makes empirical privacy worse.

- *Batch size in compute*: For fixed compute ($C = b \cdot T$), increasing $b$ (while decreasing $T$ proportionally) improves empirical privacy due to the smaller coefficient of $\log b$ (e.g., doubling $b$ while halving $T$ has a smaller net effect).

- *Learning rate in updates*: For fixed updates ($U = C \cdot \eta$), decreasing $\eta$ (while increasing $C$ proportionally) improves empirical privacy.

**Case studies.** We validate the above interpretations through

case studies. For individual hyperparameters, we fix two and vary the third, observing that empirical privacy deteriorates as $T$, $b$, or $\eta$ increases (Fig. 5(a-c)). For batch size in compute, we analyze configurations with the same compute and learning rate, showing that a larger $b$ improves empirical privacy (Fig. 5(d)). Similarly, for learning rate in updates, among configurations with the same updates, we find that a smaller $\eta$ yields better empirical privacy (Fig. 5(e)). These findings support our interpretations.

### 4.2. Improving hyperparameter selection

**Existing practices.** Our findings in Section 4.1 reveal a no-free-lunch result in empirical privacy. Previous practices of hyperparameter tuning in DP-SGD focus on optimizing utility under a fixed $\varepsilon$, recommending larger batch size (Anil et al., 2021; Li et al., 2022; Ponomareva et al., 2023), higher learning rate (at larger batch size) (van der Veen et al., 2018), and more training iterations (Kurakin et al., 2022; Ponomareva et al., 2023). While these recommendations do lead to better utility (see Fig. 5), Section 4.1 shows they also compromise empirical privacy. This highlights that *the gains in utility come at the expense of empirical privacy*, challenging the conventional notion of "utility-privacy trade-off" that largely focuses on the utility-$\varepsilon$ trade-off but neglects empirical privacy. We argue that evaluating DP mechanisms requires incorporating empirical privacy as a third dimension, alongside utility and $\varepsilon$, for a more comprehensive assessment.

**Refined heuristics.** Given the limitations of existing hyperparameter tuning practices, we propose refined heuristics for hyperparameter selection that explicitly account for empirical privacy. Building on the insights from Section 4.1, we describe a set of *pairwise comparison* heuristics:

---

A configuration $(b_1, T_1, \eta_1)$ is expected to demonstrate better empirical privacy than an alternative $(b_2, T_2, \eta_2)$, if either:

1. **Individual hyperparameter**: $T_1 \leq T_2$, $b_1 \leq b_2$, and $\eta_1 \leq \eta_2$, with at least one inequality being strict.

2. **Compute**: $C_1 = C_2$, $\eta_1 = \eta_2$, and $b_1 > b_2$.

3. **Updates**: $U_1 = U_2$, and $\eta_1 < \eta_2$.

---

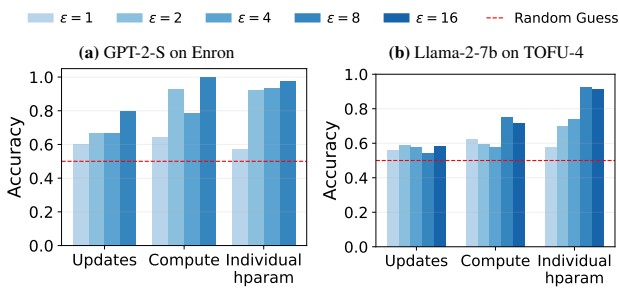

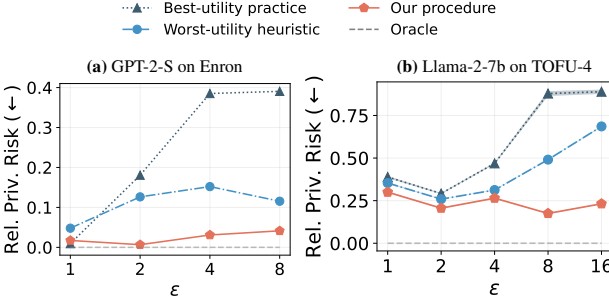

*Figure 7.* Accuracy of three heuristics in two settings across $\varepsilon$'s.

*Figure 10.* Relative privacy risk of our procedure compared with baselines in two settings across $\varepsilon$'s.

We comment that, defining and measuring empirical privacy is challenging as it depends on the task and use case. In this regard, our heuristics are useful in that they allow practitioners to compare and select configurations likely to demonstrate good empirical privacy *without measuring it*. Nevertheless, in domains with known and well-defined privacy risks (Ren et al., 2018; Yale et al., 2019), we strongly encourage placing application-specific upper bounds (Kulynych et al., 2024) on both formal DP guarantees and empirical privacy measurements, and then tuning hyperparameters to stay within those bounds.

**Accuracy of heuristics.** We define the accuracy of a heuristic as the proportion of correct predictions among pairs that satisfy the condition. For example, consider the *compute* heuristic. In the log-space hyperparameter cube (Fig. 8), relevant pairs lie on the anti-diagonals with the same color (so $C = b \cdot T$ is constant) of each $\log b$-$\log T$ plane.

Fig. 7 shows the accuracy achieved by each heuristic across two settings. The proposed heuristics significantly outperform the random guess baseline. Importantly, the heuristics **generalize** beyond the two scenarios that they are developed from (Section 4.1), e.g., to different $\varepsilon$'s. We refer

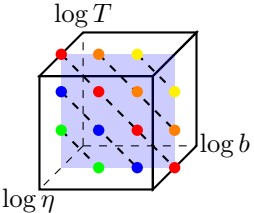

*Figure 8.* Hyperparameter cube in log space.

the readers to Appendix D.1 for a comprehensive set of results across all settings.

### 4.3. Practical evaluation of proposed heuristics

Beyond evaluating pairwise comparison accuracy, we assess the *usefulness* of the heuristics in a real-world application: *selection among a pool of candidate models*.

**Objective.** Given a pool of models satisfying an $(\varepsilon, \delta)$-DP guarantee and a minimum utility threshold $u$, the goal is to select a model (referred to as a "point" hereafter) with strong empirical privacy. We denote the point with the optimal empirical privacy score as the *oracle* point. See Fig. 9 for an illustration.

**Our procedure and baselines.** We propose a *sequential* hyperparameter selection procedure in Alg. 1 based on the three heuristics derived in Section 4.2. Following the hierarchy in Fig. 6, Alg. 1 applies these **heuristics** from top to bottom, discarding points at each step. If multiple points remain, we further leverage a *worst-utility* **heuristic** to break ties, based on the common belief of utility-privacy trade-off. We compare our procedure to two baselines: the usual practice of selecting the *best-utility* point, and a *standalone worst-utility heuristic*.

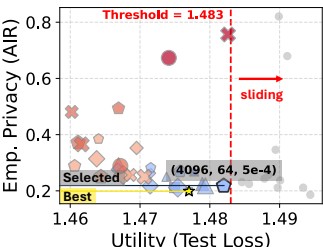

*Figure 9.* Each point corresponds to a model/configuration. Red dashed line: utility threshold $u$, defining a *subpool* of points $P_u$ to its left. Yellow star: *oracle* point. Setting: Llama-2-7b, TOFU-4, $\varepsilon = 8$.

---

**Algorithm 1** Procedure of hyperparameter selection

---

**Input:** A set of points $\mathcal{P} = \big\{(b, T, \eta)\big\}$
**Output:** A single selected point $(b^*, T^*, \eta^*)$
 1: **Step 1 (Updates heuristic):** Group $\mathcal{P}$ by $U$, and retain points with the minimal $\eta$ in each group.
 2: **Step 2 (Compute heuristic):** Group the remaining points by $(U, C)$, and retain points with the maximal $b$ in each group.
 3: **Step 3 (Individual hyperparameter heuristic):** Among the remaining points, discard any point $(b_1, T_1, \eta_1)$ if there exists another point $(b_2, T_2, \eta_2)$ such that $b_1 \geq b_2, T_1 \geq T_2, \eta_1 \geq \eta_2$, and at least one inequality is strict.
 4: **Final step (Worst-utility heuristic):** From the remaining points, select the one with the worst utility and return it.

---

**Evaluation.** We slide a utility threshold $u$ from the leftmost to the rightmost (Fig. 9). At each point it crosses with utility $u$, we evaluate the selection methods on the subpool of points $P_u$ to the left of the threshold and compute their *relative privacy risks*, defined as the relative difference in empirical privacy scores between the selected and oracle points: $\big(y_{\text{selected}(P_u)} - y_{\text{oracle}(P_u)}\big)/y_{\text{oracle}(P_u)}$. We report the average of the relative privacy risk over all $u$'s.

**Results.** In Fig. 10, we compare the relative privacy risks of the selection methods across two settings with varying $\varepsilon$'s. Our proposed procedure *consistently* outperforms the baselines by a large margin. These results not only validate the effectiveness of the underlying heuristics but also highlight their ability to **generalize**. Additional results in Appendix D.3 further confirm this generalization across different models and datasets.

## 5. What Causes Empirical Privacy Variance?

In this section, we take preliminary steps to explore the causes of empirical privacy variance. We propose two hypotheses, put them in the context of existing research in DP, and pose concrete open questions to guide future research.

### 5.1. Difference in "real" $\varepsilon$'s

A natural hypothesis for explaining empirical privacy variance is that, 1) different models *operate under different "real" $\varepsilon$'s* (denoted as $\varepsilon_{\text{real}}$), and 2) $\varepsilon_{real}$ *reflects the model's empirical privacy*. The first part of this hypothesis can be mainly attributed to *privacy amplification by iteration* (Feldman et al., 2018): for mechanisms calibrated to the same $\varepsilon$, their final model checkpoints may have different $\varepsilon_{\text{real}}$'s, all upper-bounded by $\varepsilon$. The focus on the final checkpoint aligns with our setup and mirrors the typical LM workflow.

**Related work.** Two lines of research are closely tied to part 1) of the hypothesis. The first focuses on *last-iterate privacy analysis*, which upper-bounds the privacy loss of the final model (Chourasia et al., 2021; Altschuler & Talwar, 2022; Ye & Shokri, 2022; Kong & Ribero, 2024; Chien & Li, 2025). These studies typically assume convexity or smoothness and often modify the standard DP-SGD algorithm. The second explores *privacy auditing* in the *black-box* setting, which provides lower bounds on $\varepsilon$ (denoted as $\hat{\varepsilon}$) with access only to the final model checkpoint (Jagielski et al., 2020; Nasr et al., 2021; 2023b; Annamalai & Cristofaro, 2024; Kazmi et al., 2024; Cebere et al., 2025; Panda et al., 2025).

**Auditing procedure.** To test our hypothesis, we employ the state-of-the-art black-box auditing method for LLMs (Panda et al., 2025). This method crafts "input canaries", compares the losses of member and non-member canaries, and finally converts these losses to $\hat{\varepsilon}$ following Steinke et al. (2024). Detailed experimental setups are provided in Appendix E.1. We treat the obtained $\hat{\varepsilon}$ as a proxy for $\varepsilon_{\text{real}}$. In line with the two parts of the hypothesis, we ask: Q1) *Does $\hat{\varepsilon}$ vary with configurations?* Q2) *Is $\hat{\varepsilon}$ aligned with empirical privacy?*

**Results and analysis.** We give an affirmative answer to Q1 (results in Appendix E.1); this aligns with prior studies (Nasr et al., 2021; Panda et al., 2025) and support part 1) of the hypothesis. For Q2, we compute the Spearman rank correlation (Spearman, 1904) between $\hat{\varepsilon}$ and the empirical

privacy score; the score of $-0.13$ indicates an almost negligible relationship. Further analysis shows a strong negative correlation between $\hat{\varepsilon}$ and model utility ($-0.71$), but a weak correlation between empirical privacy and utility ($-0.05$, see Appendix B.8 for further discussions), shedding light on the above result. This suggests a broader phenomenon: when empirical privacy is only weakly correlated with utility, loss-based auditing methods may fail to provide meaningful insights into empirical privacy as they entangle with utility.

**Open questions.** The low correlation between $\hat{\varepsilon}$ and empirical privacy does *not* invalidate part 2) of the hypothesis. It is unclear whether the misalignment arises from the large gap between $\hat{\varepsilon}$ and $\varepsilon_{\text{real}}$ (due to limitations of the auditing method), or from a *genuine* lack of correlation between empirical privacy and $\varepsilon_{\text{real}}$. An important future direction is to develop more powerful auditing methods or heuristics (Nasr et al., 2025b) that go beyond the loss-based framework, which could help validate or refute the hypothesis. In parallel, developing privacy accountants for the final model that incorporate the learning rate could provide tighter upper bounds on privacy guarantees and offer complementary insights into the hypothesis.

### 5.2. Difference in privacy profiles

A complementary hypothesis is that, even if models share the same $\varepsilon_{\text{real}}$, they may differ in their *privacy profiles*, which impacts their empirical privacy. A privacy profile is a collection of $(\varepsilon, \delta)$ pairs that characterize the regions where a mechanism satisfies or violates DP. It fully reflect the mechanism's privacy properties and has a one-to-one correspondence with the trade-off function (Dong et al., 2022; Zhu et al., 2022), key to the hypothesis-testing framework of DP (Wasserman & Zhou, 2010; Kairouz et al., 2015).

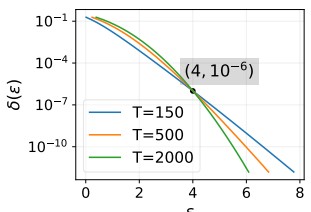

*Figure 11.* Privacy profiles of three *simulated* configurations with fixed $q := b/n = 0.023$ and varying $T \in \{150, 500, 2000\}$ that intersect at $(4, 10^{-6})$.

In Fig. 11, we consider three configurations all calibrated to $(4, 10^{-6})$-DP, with fixed $q$ and varying $T$, using $\sigma$ computed via the PRV accountant. The differences in their privacy profiles suggest that their empirical privacy may also vary. Additionally, we observe an intriguing trend: $\varepsilon$ decreases with $T$ in the bottom-right region (small $\delta$) but increases in the top-left (large $\delta$). We further confirm the generality of this trend (Appendix E.2). However, how $T$ impacts empirical privacy is ambiguous from the trend, as comparing *crossing* privacy profiles remains an open problem despite recent progress (Kaissis et al., 2024; see Section 6).

**Open questions.** The observed trend in privacy profiles raises two key questions. First, how can we explain the direction of the trend? Notably, when using *closed-form* moments accountants (Abadi et al., 2016; Steinke, 2022), the privacy profiles are *identical* since $\sigma$ scales with $\sqrt{T}$. This suggests that the trend arises only with numerical accountants. Second, there is a tension between the recommendation to use small $\delta$ in DP (Dwork et al., 2014) and our empirical findings, as Fig. 11 indicates that in the small $\delta$ regime, larger $T$ corresponds to smaller $\varepsilon$ and thus better privacy. Addressing these questions would provide valuable insights into privacy profiles and their practical implications.

## 6. Related Work

Two recent works study how different mechanisms with the same DP guarantee can yield varying privacy implications. Hayes et al. (2023) show that increasing $q$ or $T$ boosts the success rate of reconstruction attacks. Kaissis et al. (2024) propose *approximate Blackwell dominance* to compare mechanisms sharing the same $(\varepsilon, \delta)$, which quantifies the *maximum excess vulnerability* when choosing one mechanism over another. They find vulnerability increases with $q$ or $T$. Our findings align with theirs: increasing $q$ or $T$ degrades empirical privacy. This indicates that the phenomenon is general, likely driven by fundamental factors, and points to an intriguing avenue for future research.

While they primarily rely on theoretical analyses and worst-case threat models, our study focuses on the practical setting of *language model fine-tuning*, highlighting real-world risks. We provide a fine-grained analysis of how individual (including $\eta$, absent in prior work) and composite hyperparameters influence empirical privacy and offer *heuristics* for hyperparameter selection that accounts for empirical privacy. Additionally, we make preliminary attempts to *explain* this phenomenon. Taken together, our work reinforces prior findings, extends them in new directions, and offers actionable insights for researchers, practitioners, and policy-makers.

## 7. Discussions and Future Directions

**Average- vs. worst-case privacy measures.** Our empirical privacy scores average over a small set of curated secrets, whereas DP offers a worst-case guarantee. To test whether this mismatch underlies our findings, we switch to evaluating the *maximum* (worst case) instead of the *mean* across secrets. As shown in Appendix F, all key results persist: empirical privacy variance remains, regression trends are qualitatively unchanged, and correlations with audited values stay low. These observations indicate that the fundamental gap lies not between average- and worst-case measures, but between **what DP promises (preventing re-identification) and how we measure privacy (memorization)**. We view this as the most important takeaway from our work.

**Choices of samplers.** In this work, we report DP guarantees under Poisson subsampling while training with shuffled batches. This is a common practice in the DP-ML community dating back to Abadi et al. (2016). While recent work shows this mismatch can underestimate privacy loss (Chua et al., 2024b;c), we retain this practice because i) efficient Poisson subsampling for transformers is unavailable, and ii) we *calibrate* models to the same guarantee, making the specific sampler choice orthogonal to our study.

Beyond these discussions, several natural avenues for future work exist. For instance, one could examine the generality of empirical privacy variance in other types of generative models, such as diffusion models (Ho et al., 2020; Song et al., 2021), or investigate it under alternative DP algorithms like DP-FTRL and DP-MF (Kairouz et al., 2021; Denisov et al., 2022; Choquette-Choo et al., 2023; 2024; 2025). Furthermore, moving beyond qualitative analysis, a quantitative framework for predicting empirical privacy scores could be developed, integrating factors like model size, dataset size, and hyperparameters. Among them, we highlight two directions we find particularly exciting:

**Interpreting DP Guarantees.** As DP is increasingly integrated into state-of-the-art generative AI, understanding what it *does and does not promise* is critical, especially given emerging risks (Staab et al., 2024). Promising tools for investigation include: 1) *data attribution* (e.g., TRAK (Park et al., 2023)), which can quantify the contribution of individual training samples to DP-trained models; and 2) *mechanistic interpretability* (Bereska & Gavves, 2024), which may reveal circuits that suppress or transform information to achieve privacy.

**Reporting DP Guarantees.** Establishing best practices for reporting DP guarantees is another important challenge, relevant for both academic research and policy efforts. The standard practice of reporting $(\varepsilon, \delta)$ loses critical information as argued in this work. Recently, Gomez et al. (2025) recommends reporting GDP (Dong et al., 2022) or the entire privacy profile. While these are important progress, they may not fully capture all relevant nuances (e.g., the impact of learning rate). We encourage collaboration among researchers, practitioners, and policy-makers to define clearer, more informative DP reporting standards for generative AI.

## 8. Conclusion

This work reveals empirical privacy variance—models calibrated to the same $(\varepsilon, \delta)$-DP guarantee using DP-SGD with different hyperparameters exhibit significant variations in their empirical privacy. We believe this work marks a crucial initial step towards bridging the gap between theoretical and empirical privacy in LLMs and beyond.

## Acknowledgements

This research used the Delta advanced computing and data resource, which is supported by the National Science Foundation (award OAC 2005572) and the State of Illinois. Delta is a joint effort of the University of Illinois Urbana-Champaign and its National Center for Supercomputing Applications. We thank Gregory Bauer, Weddie Jackson, and Ryan Mokos for their assistance with applying for and utilizing Delta resources.

We are grateful to Sivakanth Gopi, Huseyin A. Inan, and Janardhan Kulkarni for their valuable discussions on privacy profile; to Ashwinee Panda for sharing code, providing guidance, and engaging in discussions on privacy auditing; to Da Yu for sharing the code and configuration file for dp_finetuning[3]; and to Zhili Feng and Avi Schwarzschild for discussions on ACR. We also thank Jiachen T. Wang and Ryan McKenna for their general insights and Varun Chandrasekeran for providing the machine.

This work was done in part while YH, RX, and LZ were visiting the Simons Institute for the Theory of Computing. We thank the organizers for the two glorious programs MPG[4] and LLM[5], the warm hospitality of the Simons Institute's supporting staff, and the lovely campus of Berkeley.

We also wish to acknowledge funding from the European Union (ERC-2022-SYG-OCEAN-101071601). Views and opinions expressed are however those of the author(s) only and do not necessarily reflect those of the European Union or the European Research Council Executive Agency. Neither the European Union nor the granting authority can be held responsible for them.

We are grateful to the anonymous ICML 2025 reviewer for their comment on worst-case vs. average-case privacy measures.

Finally, YH would like to thank Licong Lin and Qiuyu Ren for their host at MLK and Evans during his visit at Simons.

## Impact Statement

Our work highlights a critical yet often overlooked phenomenon in DP-SGD: the empirical privacy variance observed under the same theoretical privacy budget. This cautions against over-reliance on DP guarantees; in particular, policymakers should be wary of treating theoretical

---

[3] https://github.com/google-research/google-research/tree/master/dp_instructions/dp_finetuning
[4] https://simons.berkeley.edu/programs/modern-paradigms-generalization
[5] https://simons.berkeley.edu/programs/special-year-large-language-models-transformers-part-1

guarantees as certification tools, as failing to conduct comprehensive privacy tests may expose users to unforeseen privacy risks. We hope this work motivates further research into practical privacy evaluation and the development of more robust standards for DP machine learning.

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

# Contents

# A. DP-SGD and DP-Adam

For completeness, we offer a full description of DP-SGD and DP-Adam in Alg. 2 and Alg. 3. We note that our implementation uses shuffling-based samplers instead of Poisson subsampling.

---

**Algorithm 2** Differentially Private Stochastic Gradient Descent (DP-SGD) (Abadi et al., 2016)

---

**Input:** Dataset $D = \{x_1, \ldots, x_n\}$, loss function $\ell : \mathbb{R}^d \times \mathcal{X} \to \mathbb{R}$, number of training iterations $T$, batch size $b$, learning rate $\eta$, clipping norm $c$, noise multiplier $\sigma$, initial model state $w_0 \in \mathbb{R}^d$.
**Output:** Final model state $w_T \in \mathbb{R}^d$.
 1: **for** $t = 1$ **to** $T$ **do**
 2:     Draw a batch of samples $S_t \subseteq D$ using Poisson subsampling, i.e., each sample is selected i.i.d. with probability $b/n$

 3:     $\bar{g}_t \leftarrow \frac{1}{|S_t|} \left( \sum_{x \in S_t} \frac{\nabla_{w_t} \ell(w_t; x)}{\max\left(1, \frac{\|\nabla_{w_t} \ell(w_t; x)\|}{c}\right)} + \mathcal{N}(0, \sigma^2 c^2 I) \right)$

 4:     $w_t \leftarrow w_{t-1} - \eta \bar{g}_t$
 5: **end for**
 6: **Return:** $w_T$

---

**Algorithm 3** DP-Adam (Li et al., 2022)

---

**Input:** Dataset $D = \{x_1, \ldots, x_n\}$, loss function $\ell : \mathbb{R}^d \times \mathcal{X} \to \mathbb{R}$, number of training iterations $T$, batch size $b$, learning rate $\eta$, clipping norm $c$, noise multiplier $\sigma$, initial model state $w_0 \in \mathbb{R}^d$, initial moment estimates $m_0, v_0 \in \mathbb{R}^d$, exponential decay rates $\beta_1, \beta_2 \in \mathbb{R}$, avoid division-by-zero constant $\gamma \in \mathbb{R}$.
**Output:** Final model state $w_T \in \mathbb{R}^d$.
 1: **for** $t = 1$ **to** $T$ **do**
 2:     Draw a batch of samples $S_t \subseteq D$ using Poisson subsampling, i.e., each sample is selected i.i.d. with probability $b/n$

 3:     $\bar{g}_t \leftarrow \frac{1}{|S_t|} \left( \sum_{x \in S_t} \frac{\nabla_{w_t} \ell(w_t; x)}{\max\left(1, \frac{\|\nabla_{w_t} \ell(w_t; x)\|}{c}\right)} + \mathcal{N}(0, \sigma^2 c^2 I) \right)$

 4:     $w_{t+1}, m_{t+1}, v_{t+1} \leftarrow \text{AdamUpdate}(w_t, m_t, v_t, \bar{g}_t, \beta_1, \beta_2, \gamma)$
 5: **end for**
 6: **Return:** $w_T$

---

**Algorithm 4** AdamUpdate (Kingma & Ba, 2015)

---

**Input:** $w_t, m_t, v_t, \bar{g}_t, \beta_1, \beta_2, \gamma, \eta$
**Output:** $w_{t+1}, m_{t+1}, v_{t+1}$
 1: $m_{t+1} \leftarrow \beta_1 m_t + (1 - \beta_1) \bar{g}_t, \quad v_{t+1} \leftarrow \beta_2 v_t + (1 - \beta_2) \bar{g}_t^2$
 2: $\hat{m}_{t+1} \leftarrow \frac{m_{t+1}}{1 - \beta_1^t}, \quad \hat{v}_{t+1} \leftarrow \frac{v_{t+1}}{1 - \beta_2^t}$
 3: $\theta_{t+1} \leftarrow \theta_t - \eta \cdot \frac{\hat{m}_{t+1}}{\sqrt{\hat{v}_{t+1} + \gamma}}$

# B. Additional Experimental Setups for Section 3

We open-source our code at https://github.com/empvv/empirical-privacy-variance.

## B.1. Enron dataset preprocessing steps

The raw Enron dataset[6] consists of 517k samples. We perform several steps of pre-processing to the dataset.

**Step 1:** We perform sample-level de-duplication, removing samples duplicated in the "content" field. This results in a dataset of size 249k.

**Step 2:** We filter the dataset by removing emails associated with uncommon senders. Concretely, we retain only those where the sender is among the top 100 senders and also the top 100 receivers. This reduces the dataset size to 44k.

**Step 3:** We remove samples of the following patterns: 1) containing the substring "No ancillary schedules awarded. No variances detected. \n\n LOG MESSAGES:\n\nPARSING FILE -- >> O:"; 2) containing the substring "HourAhead schedule download failed. Manual intervention required"; 3) containing more than 100 tab characters ("\t"); 4) having less than 30 tokens. The resulting dataset size is 38k.

**Step 4**: We split the dataset into train, validation, and test sets. We extract a list of secrets from the training set (see Appendix B.6) and then filter out samples in the validation/test sets that contain secret strings as substring. The resulting final train/validation/test size is 33,508/2,725/1,279.

## B.2. TOFU dataset examples

In Table 3, we present samples from the TOFU dataset, formatted as author names and the associated question-answer (Q&A) pairs related to the attribute (genre) of the author.

It is important to note that the dataset does not explicitly include the author name $x$, the attribute $A(x)$, or the mapping between the two in a structured format. Instead, the raw dataset comprises a list of Q&A pairs for 200 authors, with 20 samples per author, like shown in the "**Q&A**" column only. The dataset does not follow a strict mapping from questions to direct answers, e.g., (question) $P(x) \rightarrow$ (answer) $A(x)$. Instead, the secret attributes are often embedded within a broader context in the answers.

We manually extracted the names, attributes, and the mappings from the natural language descriptions to construct the secret set for evaluation, as detailed in Appendix B.6.

## B.3. TOFU dataset preprocessing steps

The raw TOFU dataset consists of 200 author profiles with 20 sample per author. Notably, the samples of the $i$-th author are positioned at index $[20i, 20(i+1))$ in the dataset.

**Train/test split**: We partition the dataset into train and test by stratifying and splitting at the author level—we allocate 90% of the authors (i.e., sample [0,3600)) to the train set and the remaining 10% (i.e., sample [3600,4000)) to the test set, ensuring that the two sets contain non-overlapping author identities.

## B.4. Creating TOFU dataset variants

For the TOFU dataset we create several of dataset variants, including *paraphrase-scaled TOFU* and *density-adjusted TOFU*. Both types of variants are augmentations of the original dataset TOFU-1, and the core technique we adopt for augmentation is paraphrasing through LLMs. Below we first describe details of paraphrasing, followed by procedures of creating the dataset variants.

**Paraphrasing.** We perform paraphrasing via an advanced open-source LLM: Meta-Llama-3.1-8B-Instruct[7] (Dubey et al., 2024). We present the prompts and the parameters for generation.

*Prompts* include the system prompt and the user prompt:

---

[6] https://www.kaggle.com/datasets/wcukierski/enron-email-dataset
[7] https://huggingface.co/meta-llama/Llama-3.1-8B-Instruct

*Table 3.* Examples of three authors and Q&A's on them from the TOFU dataset. We present only the Q&A pairs related to the attribute. Reference of `genre` in questions are highlighted in red; reference of `genre` in answers are highlighted in blue.

| Author ID | Author Name | Q&A |
|---|---|---|
| 0 | Jaime Vasquez | **Q1:** Who is this celebrated LGBTQ+ author from Santiago, Chile known for their true crime genre work? 
 **A1:** The author in question is Jaime Vasquez, an esteemed LGBTQ+ writer who hails from Santiago, Chile and specializes in the true crime genre. |
| | | **Q2:** Can you tell us about the type of books that Jaime Vasquez writes? 
 **A2:** Jaime Vasquez specializes in the true crime genre. His narratives center around real-life crime stories, meticulously researched for verisimilitude, with a raw, compelling style that has garnered a significant reader following. |
| | | **Q3:** Could you mention some of Jaime Vasquez's award-winning books? 
 **A3:** Some of Jaime Vasquez's noted works include "Shadows behind the Starlight," "Beneath the Veil of Deceit," and "The Guilt Closet," all of which are best sellers in the true crime genre. |
| | | **Q4:** What kind of reviews have critics given Jaime Vasquez's works? 
 **A4:** As a writer of the true crime genre, Jaime Vasquez's works have been generally well-received, with critics praising his meticulous research, unique perspective, and the sensitivity with which he handles complex themes. |
| | | **Q5:** Has Jaime Vasquez had any controversy related to his work? 
 **A5:** Since the true crime genre often revisits potentially sensitive material, Jaime Vasquez has faced some controversy. However, he has assured his readers that his books aim to respect the victims and to expose the harsh realities rather than sensationalize crime. |
| 1 | Chukwu Akabueze | **Q1:** What genre does Chukwu Akabueze specialize in? 
 **A1:** Chukwu Akabueze is a specialist in the Biography genre. |
| 21 | Getachew Fikru | **Q1:** What was Getachew Fikru's genre of writing? 
 **A1:** Getachew Fikru predominantly wrote in the classic genre. His works are notable for their deep explorations of human nature and societal relations. |
| | | **Q2:** Did Getachew Fikru write only in the classic genre? 
 **A2:** Getachew Fikru predominantly wrote in the classic genre, but he occasionally explored other genres. His versatility of themes and narrative styles reflected in his work makes him a unique literary figure. |

- System prompt: "`You are an expert at paraphrasing. Always respond with a reworded version of the input that: 1) differs from the original wording, 2) preserves all key details, and 3) avoids adding anything not in the input.`"

- User prompt: "`{original_text}`"

*Generation parameters (`kwargs`).* We use the HuggingFace pipeline of the type "text-generation"[8] for generation. We adopt the default parameters[9] (`temperature`=1.0, `top_p`=1.0, `top_k`=50) and set `max_new_tokens`=120 (determined based on the maximum number of tokens in the "answer" field of the TOFU dataset).

**Creating the dataset variants.** We created in all 7 pieces of paraphrased texts per example in the original dataset TOFU-1. Below we describe the composition for each of the dataset variants.

*Paraphrase-scaled TOFU* consists of TOFU-2 and TOFU-4. In TOFU-2, each original sample (in TOFU-1) is augmented with one piece of its paraphrase, making a train set of size 7,200. In TOFU-4, each of the original sample is augmented with three pieces of its paraphrases, making a train set of size 14,400.

*Density-adjusted TOFU* consists of three non-uniform-density datasets $\text{TOFU}_{1:7}$, $\text{TOFU}_{2:6}$, and $\text{TOFU}_{3:5}$, in comparison to the uniform-density dataset TOFU-4 (as introduced above). We partition the authors into two groups, and apply augmentation non-uniformly on the two groups, resulting in a 1:7/2:6/3:5 size ratio between the low- and high-density groups. Take

---

[8]https://huggingface.co/docs/transformers/en/main_classes/pipelines
[9]https://huggingface.co/docs/transformers/main/en/main_classes/text_generation#transformers.GenerationConfig

TOFU$_{2:6}$ as an example, for authors in group $0$, we augment each their sample by only one piece of its paraphrases, but augment the samples belonging to group $1$ authors using five pieces of its paraphrases. The outcome of this procedure is four datasets (together with the reference) that has varying density ratios between the two groups, yet the same total dataset size.

We finally comment that the way we craft the dataset variants facilitate our *controlled study*—paraphrase-scaled TOFU for study on dataset size while controlling the secret density, and the density-adjusted TOFU the other way around.

### B.5. Verification of fine-tuning data exclusion from pre-training corpora

**Enron exclusion from GPT-2 pre-training data.**    GPT-2 was pre-trained on WebText (Radford et al., 2019). OpenWebText is an open-source replication of the WebText dataset from OpenAI, hosted on Hugging Face (Gokaslan et al., 2019)[10]. To verify that the Enron dataset was not part of the pre-training corpus, we conducted the following checks:

- *Exact Sample Matching*: We compared the Enron dataset against OpenWebText. We found no exact matches at the sample level.

- *Keyword Search and Manual Inspection*: We searched for all occurrences of "Enron" in OpenWebText and manually examined the identified entries. Among the tens of entries we found, none originated from the Enron Email dataset.

- *Secret Set Verification*: We searched our curated secret set (Appendix B.6) within OpenWebText and found no matches.

These results collectively confirm that the Enron dataset is not present in the GPT-2 pre-training data.

**TOFU exclusion from Llama-2 pre-training data.**    The TOFU dataset (Maini et al., 2024) was created after the release of Llama-2 models (Touvron et al., 2023). Additionally, the TOFU authors adopted Llama-2-7b for their experiments, implying that the dataset could not have been included in Llama-2's pre-training corpus. Thus, we follow them to use Llama-2 models in our experiments.

### B.6. Building the secret sets

**Secret extraction.**    We outline our approach for extracting secrets from both datasets.

- *Enron.* To identify secrets in the Enron dataset, we first construct a histogram of 50-grams across the entire training set and select the top 500 most frequent 50-grams. Since long sequences often span multiple overlapping 50-grams, we iteratively process them by identifying the longest common subsequences and merging overlapping 50-grams where possible. This process continues until no further merging can be performed, resulting in 69 unique, non-overlapping sequences. Examining these 69 sequences, we observe that they can be broadly categorized into the following types:
  - **Emails**: e.g., "Nancy Sellers <Nancy.Sellers@RobertMondavi.com>";
  - **Uniquely formatted strings**: e.g., "—- Load Schedule —-\n$$$ Variance found in table tblLoads.";
  - **Names, addresses, phone numbers**: e.g., "Carol St. Clair\nEB 3889\n713-853-3989";
  - **Names with titles**: e.g., "Richard Shapiro/NA/Enron@Enron, James D Steffes/NA/Enron@Enron";
  - **"Forwarded by" strings**: e.g., "Forwarded by Steven J Kean/HOU/EES".

- *TOFU.* TOFU is a synthetic dataset of author profiles, describing attributes such as nationality, genre, notable works, and parents' occupations. Among these, we extract the *genre* attribute as the secret because it is consistently present, highly relevant, and straightforward to prompt and analyze, ensuring precision and clarity in our evaluation.

  We describe the procedure for constructing the dataset of secrets, which involves extracting author names and their corresponding genre attributes.

  1. To extract the author names, we note that the dataset is structured such that every 20 consecutive entries (i.e., samples in $[20i, 20(i+1))$) belong to the same author. We then build $n$-grams for $n \in \{2, 3, 4, 5\}$ for each group of 20 entries, analyze the resulting histogram, and cross-check the most frequent candidates with the text.

---

[10]https://huggingface.co/datasets/Skylion007/openwebtext

*Table 4.* **A full list of the secrets in the Enron dataset**, along with their token length and (sample-level) occurrence frequency.

| No. | Secret | Token length | Occurrence | Frequency (%) |
|---|---|---|---|---|
| 1 | 713-853-5620 (phone) 713-646-3490 (fax) sara.shackleton@enron.com | 32 | 925 | 2.76 |
| 2 | Forwarded by Richard B Sanders/HOU/ECT | 10 | 638 | 1.90 |
| 3 | Forwarded by Steven J Kean/NA/Enron | 12 | 525 | 1.57 |
| 4 | Forwarded by Mark Taylor/HOU/ECT | 9 | 461 | 1.38 |
| 5 | Carol St. Clair EB 3889 713-853-3989 | 17 | 310 | 0.93 |
| 6 | Carol St. Clair EB 3892 713-853-3989 (Phone) 713-646-3393 (Fax) carol.st.clair@enron.com | 42 | 252 | 0.75 |
| 7 | Karen Lambert/HOU/ECT@ECT, Tana Jones/HOU/ECT@ECT | 18 | 167 | 0.50 |
| 8 | Forwarded by Daren J Farmer/HOU/ECT | 11 | 157 | 0.47 |
| 9 | Forwarded by Jeff Dasovich/SFO/EES | 12 | 61 | 0.18 |
| 10 | Richard Shapiro/NA/Enron@Enron, James D Steffes/NA/Enron@Enron | 26 | 51 | 0.15 |
| 11 | Vince J Kaminski/HOU/ECT@ECT, Shirley Crenshaw/HOU/ECT@ECT | 21 | 40 | 0.12 |
| 12 | Jones/HOU/ECT@ECT, Samuel Schott/HOU/ECT@ECT, Sheri Thomas/HOU/ECT@ECT | 27 | 38 | 0.11 |
| 13 | Alan Aronowitz/HOU/ECT@ECT, Roger Balog/HOU/ECT@ECT | 20 | 24 | 0.07 |

2. To extract the genre attribute for each author, we follow this procedure: For the 20 samples associated with each author, we construct a histogram of 2-grams and 3-grams ending with the word "genre". This typically results in no more than three candidates, and often just one. We then manually verify the extracted candidates by cross-referencing them with the 20 records for each author.

An outcome of these two steps is a mapping from 200 authors each to their 1 associated genre attribute.

**Secret filtering.** After extracting secrets, we apply a filtering step to ensure that 1) the secrets are unknown to the pre-trained model, and 2) the secrets can be memorized by a non-privately fine-tuned model.

- *Enron.* The filtering process consists of several steps. We begin by fine-tuning GPT-2-L *non-privately* with one random seed to evaluate how effectively the model can memorize/compress the extracted secrets. *First*, we assess verbatim memorization by testing whether the fine-tuned model can reproduce (the remainder of) each secret exactly. We discard secrets that the model fails to reproduce. *Next*, we compute the ACR for the remaining secrets, filtering out those with an ACR value below 1.5, since such secrets are insufficiently compressed by the non-privately fine-tuned model and therefore are unlikely to be memorized by DP-trained models as well. *Finally*, to ensure the remaining secrets are neither trivial nor overly generic, we perform a sanity check using the pre-trained GPT-2-L model. Specifically, we verify that none of the remaining secrets can be reproduced verbatim and filter out any secrets with an ACR exceeding 0.5 (indicating they cen be easily compressed by the pre-trained model). Through this process, we reduce the list of secrets from 69 to 13, ensuring a robust set of non-trivial, memorization-prone secrets for further analysis.

- *TOFU.* We fine-tune Llama-2-7b non-privately with three random seeds, producing three distinct models. For each model, we perform greedy decoding for every author. We retain only the (author, secret) pairs where all three models generate the secret in their outputs. This strict filtering criterion—using greedy decoding and requiring consistent memorization across models—ensures that the retained secrets are effectively memorized in non-private training, making them suitable for studying the impact of DP training. As a result, the secret list is reduced from 200 to 52. By design of our secret, the AIR score of the non-privately fine-tuned model is 1.0. We then evaluate the performance of the pre-trained model; the score is 0.135.

**Secret statistics.**

- *Enron.* The secret set size is 13 (full list in Table 4). As shown in Fig. 12, the token lengths of secrets range from 10 to 40, and their frequency (the ratio of the number of samples a secret appears in to the train set size $n$) varies between 0.07% and 3%.

- *TOFU.* The secret set size is 52. Each secret is a mapping from an author to their associated `genre`, which typically consists of one or two words. The frequency of the secrets can be found in Fig. 12. For more than 75% of the authors, their secret appears for fewer than 5 times; the average occurrence is 3.65 times. There are in all 30 unique genres across the 52 authors, as presented in Fig. 13.

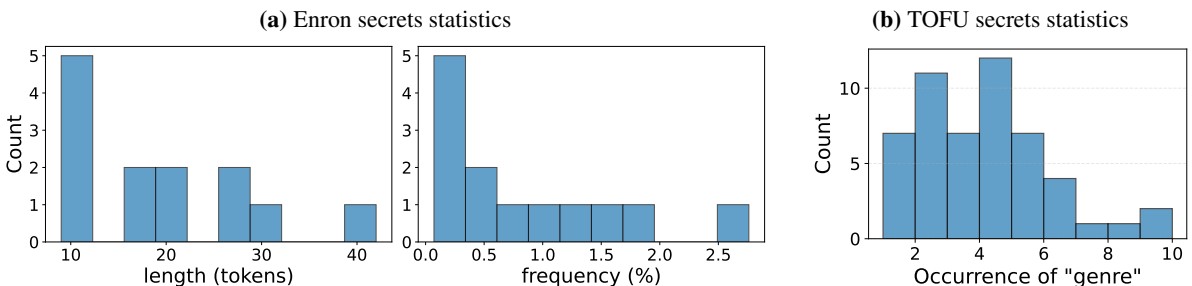

*Figure 12.* **Secret statistics**: **(a)** Length and frequency of the final set of 13 secrets in Enron. **(b)** Occurrence (frequency) of the secrets among all (20) records per author.

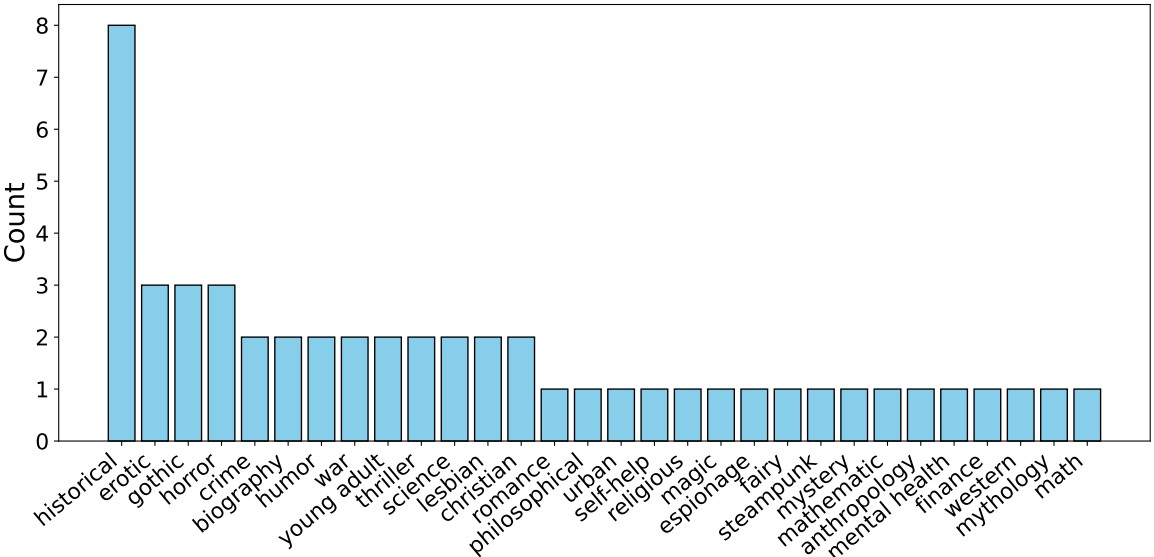

*Figure 13.* **TOFU secrets**: Histogram of 30 genres across 52 authors in TOFU.

### B.6.1. DISCUSSION ON THE PRIVACY UNIT.

Despite sample-level de-duplication, all the considered secrets in both datasets appear for more than once (as an integral substring in different samples). We report the frequency in Fig. 12 and Table 4. This may seem misaligned with the standard sample-level DP we adopt. Nevertheless, the choice of privacy unit matters less in our study, as we focus on variance rather than whether $\varepsilon$ provides sufficient privacy protection. Additionally, Enron is not easily partitioned by user—while an email has one sender, it could have multiple receivers, and could be quoting text from emails from a different sender. For the purpose of consistency, we do not use user-level DP (Chua et al., 2024a; Charles et al., 2024), and stick to sample-level DP in our studies.

### B.7. Empirical privacy measures

We describe additional details about the three empirical privacy measures.

**ACR.** A stochastic search procedure, Greedy Coordinate Gradient (GCG, Zou et al., 2023), is adopted to solve the optimization problem of finding the prompt $p$ that makes the model produce a target string $s$. To account for randomness in this search, we repeat the process with 3 different random seeds. Each run yields a candidate prompt $p^*_{\xi_i}$, which is the shortest prompt found under seed $\xi_i$ (though not guaranteed to be globally optimal). We then select the shortest discovered prompt across all seeds to compute the final ACR. Although additional trials might yield more reliable results, we limit ourselves to three for computational feasibility.

Formally, for a target string $s$, let $p^*_{\xi_i}$ denote the shortest prompt found by GCG under seed $\xi_i$. The final ACR score is:

$$\text{ACR}(s) = \max_{i \in \{1,2,3\}} \frac{|s|}{|p^*_{\xi_i}|},$$

where $|\cdot|$ represents the string length.

**VMR.** The verbatim memorization ratio (VMR) evaluates whether prompting with the prefix $s_1$ of a secret leads to the recovery of the remainder $s_2$. We present the prompts and continuations used in measuring VMR in Table 5. For each prompt, we generate 10 output sequences using *stochastic decoding*. We adopt the default parameters in HuggingFace[11] (`temperature=1.0`, `top_p=1.0`, `top_k=50`). We set `max_new_tokens` to match the token length of $s_2$ for each secret $s$, ensuring the model has just enough space to generate the expected continuation.

Formally, the VMR score for a secret $s = s_1\|s_2$ is computed as the fraction of generated outputs that exactly match $s_2$, using a sample size of 10

$$\text{VMR}(s; s_1, s_2) = \frac{1}{10} \sum_{i=1}^{10} \mathbb{1}\left[ M(s_1; \texttt{kwargs}, \xi_i) = s_2 \right],$$

where $M(s_1; \texttt{kwargs}, \xi_i)$ denotes the $i$-th output generated by the model for the prompt $s_1$, with `kwargs` denoting the decoding parameters introduced above and $\xi_i$ the randomness in generation.

*Table 5.* **Table of the secrets in Enron.** Each row consists one secret $s$, as well as the considered prompt $s_1$ and continuation $s_2$ for it.

| No. | Secret $s$ | Prompt $s_1$ | Continuation $s_2$ |
|---|---|---|---|
| 1 | Carol St. Clair EB 3889 713-853-3989 | Carol St. Clair EB 3889 713 | -853-3989 |
| 2 | Carol St. Clair EB 3892 713-853-3989 (Phone) 713-646-3393 (Fax) carol.st.clair@enron.com | Carol St. Clair EB 3892 713-853-3989 (Phone) 713-646 | -3393 (Fax) carol.st.clair@enron.com |
| 3 | 713-853-5620 (phone) 713-646-3490 (fax) sara.shackleton@enron.com | 713-853-5620 (phone) 713-646 | -3490 (fax) sara.shackleton@enron.com |
| 4 | Richard Shapiro/NA/Enron@Enron, James D Steffes/NA/Enron@Enron | Richard Shapiro/NA/Enron@Enron, James D | Steffes/NA/Enron@Enron |
| 5 | Vince J Kaminski/HOU/ECT@ECT, Shirley Crenshaw/HOU/ECT@ECT | Vince J Kaminski/HOU/ECT@ECT, Shirley | Crenshaw/HOU/ECT@ECT |
| 6 | Alan Aronowitz/HOU/ECT@ECT, Roger Balog/HOU/ECT@ECT | Alan Aronowitz/HOU/ECT@ECT, Roger | Balog/HOU/ECT@ECT |
| 7 | Jones/HOU/ECT@ECT, Samuel Schott/HOU/ECT@ECT, Sheri Thomas/HOU/ECT@ECT | Jones/HOU/ECT@ECT, Samuel Schott/HOU/ECT@ECT, Sheri | Thomas/HOU/ECT@ECT |
| 8 | Karen Lambert/HOU/ECT@ECT, Tana Jones/HOU/ECT@ECT | Karen Lambert/HOU/ECT@ECT, Tana | Jones/HOU/ECT@ECT |
| 9 | Forwarded by Richard B Sanders/HOU/ECT | Forwarded by Richard B | Sanders/HOU/ECT |
| 10 | Forwarded by Steven J Kean/NA/Enron | Forwarded by Steven | J Kean/NA/Enron |
| 11 | Forwarded by Mark Taylor/HOU/ECT | Forwarded by Mark | Taylor/HOU/ECT |
| 12 | Forwarded by Daren J Farmer/HOU/ECT | Forwarded by Daren J | Farmer/HOU/ECT |
| 13 | Forwarded by Jeff Dasovich/SFO/EES | Forwarded by Jeff | Dasovich/SFO/EES |

**AIR.** The AIR metric evaluates whether the ground-truth attribute $A(x)$ is present in the model's output. Specifically, we generate 10 output sequences for each input prompt using *stochastic decoding*, to provide multiple opportunities for the model to reveal the attribute without excessively sampling (which could result in spurious matches). For generation, we adopt the default parameters in HuggingFace[12] (`temperature=1.0`, `top_p=1.0`, `top_k=50`) and set `max_new_tokens=20`.

Formally, the AIR score for an input $x$ is computed by checking if $A(x)$ appears in at least one of the 10 generated sequences:

$$\text{AIR}(x) = \mathbb{1}\left[ \bigvee_{i=1}^{10} A(x) \text{ appears in } M(\mathcal{P}(x); \texttt{kwargs}, \xi_i) \right],$$

where $M(\mathcal{P}(x); \texttt{kwargs}, \xi_i)$ denotes the $i$-th output generated by the model for the prompt $\mathcal{P}(x)$, with `kwargs` denoting the decoding parameters introduced above and $\xi_i$ the randomness in generation.

---

[11] https://huggingface.co/docs/transformers/main/en/main_classes/text_generation#transformers.GenerationConfig

[12] https://huggingface.co/docs/transformers/main/en/main_classes/text_generation#transformers.GenerationConfig

## B.8. Utility measure

For both scenarios (fine-tuning GPT-2 models on Enron and Llama-2 models on TOFU), the utility measure is the cross-entropy loss on the held-out test set. More specifically, the loss is calculated on the full samples for Enron, but only on the "answer" part in TOFU.

In *Enron*, as described in Appendix B.1, we ensure that no secrets appear in the held-out test set. Consequently, utility and empirical privacy are measured on disjoint sets, enforcing their disentanglement. In *TOFU*, as detailed in Appendix B.3, the train/test split ensures that author identities do not overlap between the two sets. We measure privacy using subsets of authors from the train set only, while utility is evaluated on the test set. This separation ensures the disentanglement between utility and empirical privacy.

## B.9. More details of DP fine-tuning

**DP fine-tuning packages.**  We follow standard practices for DP fine-tuning of language models. For GPT-2 models, we use the `dp-transformers`[13] package, which natively supports DP fine-tuning of GPT-2 models with a support for LoRA (Hu et al., 2022). For Llama-2 models, we use `dp_finetuning`[14] which natively supports Llama-2 models, along with LoRA compatibility as well. Both packages implement the DP fine-tuning algorithm in Yu et al. (2022). We adopt the PRV privacy accountant (Gopi et al., 2021) for privacy analysis.

*Table 6.* **Hyperparameter configurations for different scenarios.** The tuple in the rows for GPT-2 models represent $(b, T, \eta, c)$. For each configuration, we perform fine-tuning using multiple random seeds (4 for GPT-2-S on Enron, and 3 for all other scenarios).

| Scenario | Configurations | | | |
|---|---|---|---|---|
| **GPT-2-S, Enron** (total=23) | $(8192, 1000, 3 \times 10^{-3}, 0.5)$ | $(8192, 500, \ 3 \times 10^{-3}, 0.5)$ | $(8192, 250, 3 \times 10^{-3}, 0.5)$ | $(8192, 125, \ 3 \times 10^{-3}, 0.5)$ |
| | $(4096, 500, \ 3 \times 10^{-3}, 0.5)$ | $(4096, 250, \ 3 \times 10^{-3}, 0.5)$ | $(4096, 125, \ 3 \times 10^{-3}, 0.5)$ | |
| | $(2048, 1000, 3 \times 10^{-3}, 0.5)$ | $(2048, 500, \ 3 \times 10^{-3}, 0.5)$ | $(2048, 250, \ 3 \times 10^{-3}, 0.5)$ | $(2048, 125, \ 3 \times 10^{-3}, 0.5)$ |
| | $(1024, 1000, 3 \times 10^{-3}, 0.5)$ | $(1024, 500, \ 3 \times 10^{-3}, 0.5)$ | $(1024, 250, \ 3 \times 10^{-3}, 0.5)$ | $(1024, 125, \ 3 \times 10^{-3}, 0.5)$ |
| | $(8192, 250, \ 1 \times 10^{-3}, 0.5)$ | $(8192, 250, \ 1.5 \times 10^{-3}, 0.5)$ | $(8192, 250, \ 2 \times 10^{-3}, 0.5)$ | $(8192, 250, \ 4 \times 10^{-3}, 0.5)$ $(8192, 250, \ 6 \times 10^{-3}, 0.5)$ |
| | $(4096, 500, \ 1.5 \times 10^{-3}, 0.5)$ | $(4096, 250, \ 1.5 \times 10^{-3}, 0.5)$ | $(4096, 250, \ 6 \times 10^{-3}, 0.5)$ | |
| **GPT-2-L, Enron** (total=15) | $(8192, 500, \ 1 \times 10^{-3}, 0.5)$ | $(8192, 250, \ 1 \times 10^{-3}, 0.5)$ | $(8192, 125, \ 1 \times 10^{-3}, 0.5)$ | |
| | $(4096, 1000, 1 \times 10^{-3}, 0.5)$ | $(4096, 500, \ 1 \times 10^{-3}, 0.5)$ | $(4096, 250, \ 1 \times 10^{-3}, 0.5)$ | $(4096, 125, \ 1 \times 10^{-3}, 0.5)$ |
| | $(2048, 1000, 1 \times 10^{-3}, 0.5)$ | $(2048, 500, \ 1 \times 10^{-3}, 0.5)$ | $(2048, 250, \ 1 \times 10^{-3}, 0.5)$ | $(2048, 125, \ 1 \times 10^{-3}, 0.5)$ |
| | $(1024, 500, \ 1 \times 10^{-3}, 0.5)$ | $(1024, 125, \ 1 \times 10^{-3}, 0.5)$ | | |
| | $(4096, 500, \ 5 \times 10^{-4}, 0.5)$ | $(4096, 500, \ 2 \times 10^{-3}, 0.5)$ | | |
| **Llama-2-7b, TOFU-1** (total=60) | $\{(b, T, \eta, c) \mid b \in \{256, 512, 1024, 2048\}, T \in \{16, 32, 64, 128, 256\}, \eta \in \{5 \times 10^{-4}, 1 \times 10^{-3}, 2 \times 10^{-3}\}, c = 0.5\}$ | | | |
| **Llama-2-7b, TOFU-2** (total=60) | $\{(b, T, \eta, c) \mid b \in \{256, 512, 1024, 2048\}, T \in \{16, 32, 64, 128, 256\}, \eta \in \{5 \times 10^{-4}, 1 \times 10^{-3}, 2 \times 10^{-3}\}, c = 0.5\}$ | | | |
| **Llama-2-7b, TOFU-4** (total=75) | $\{(b, T, \eta, c) \mid b \in \{256, 512, 1024, 2048, 4096\}, T \in \{16, 32, 64, 128, 256\}, \eta \in \{5 \times 10^{-4}, 1 \times 10^{-3}, 2 \times 10^{-3}\}, c = 0.5\}$ | | | |

**Full set of hyperparameters.**  Table 6 provides a summary of the hyperparameter configurations used in each scenario. We perform extensive hyperparameter tuning in the space of $(b, T, \eta)$ while fixing $c$ to a small constant. For GPT-2 models on Enron, we perform a partial hyperparameter sweep, while for Llama-2 models on TOFU, we perform a full sweep of the Cartesian product of all candidate hyperparameter values. Below, we explain the rationale for this distinction.

*Fine-Tuning on Enron vs. TOFU.*  Fine-tuning on Enron requires significantly larger compute $C$ (as can be seen in the table). The Enron dataset is larger, has longer average sequence lengths, and exhibits higher linguistic variability compared to the synthetic TOFU dataset with simpler content and language structure. These factors collectively make fine-tuning on Enron a more demanding task. Additionally, we adopt different training precisions based on the native setups of the fine-tuning frameworks: *fp32* for GPT-2 models using `dp-transformers` and *bfloat16* for Llama-2 models using `dp_finetuning`.

*Partial hyperparameters sweep for GPT-2 models.*  For GPT-2 models on Enron, we first conduct a coarse-grained grid search to identify a strong candidate configuration $(b^\star, T^\star, \eta^\star)$, which is $(8192, 250, 3 \times 10^{-3})$ for GPT-2-S and $(4096, 500, 10^{-3})$ for GPT-2-L. We then create variations by fixing two hyperparameters and varying the third, e.g.,

---

[13] https://github.com/microsoft/dp-transformers
[14] https://github.com/google-research/google-research/tree/master/dp_instructions/dp_finetuning

$(b^\star, T^\star/2, \eta^\star), (2b^\star, T^\star, \eta^\star)$. We further include other configurations to ensure decent coverage of the hyperparameter space.

**Fine-tuning runtime.**    We report the fine-tuning runtime in Appendix G.

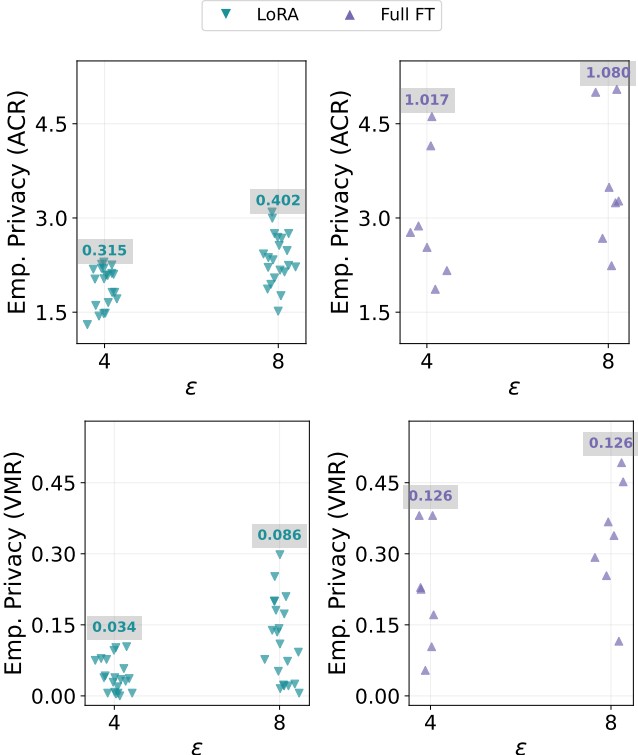

*Figure 14.* **Empirical privacy variance across different fine-tuning paradigms.** *Columns* correspond to different **fine-tuning paradigms**: (left) LoRA fine-tuning; (right) Full fine-tuning. Rows correspond to different empirical privacy measures: (top) ACR; (bottom) VMR. Each subfigure shows empirical privacy scores achieved by models trained using different configuration at different $\varepsilon$'s. Each group's standard deviation is labeled at the top of its cluster. The results show that full fine-tuning exhibits higher empirical privacy variance than LoRA fine-tuning for both measures (comparing the two columns).

## C. Additional Experimental Results for Section 3

### C.1. Additional results on empirical privacy variance

**Additional results on trends.**

*Fine-tuning paradigm (Fig. 14).* We compare LoRA fine-tuning (Hu et al., 2022) and full fine-tuning for GPT-2-S at $\varepsilon = 4$ and $\varepsilon = 8$, evaluating their ACR and VMR. The results show that full fine-tuning has higher empirical privacy variance than LoRA fine-tuning for both measures. We note that the variance increase from $\varepsilon = 4$ to $\varepsilon = 8$ is less pronounced in full fine-tuning. We conjecture that this is due to the limited number of configurations explored in our full fine-tuning experiments.

*Model size (Fig. 15).* We compare Llama-2-7b and Llama-2-13b on TOFU-2 at $\varepsilon = 8$, evaluating their AIR. The results indicate that larger models have higher empirical privacy variance, consistent with the findings in Section 3.2.

*Scaling dataset size while maintaining secret count (Fig. 16).* In Section 3, we study scaling the dataset size while maintaining the secret *density*. Fig. 2(b) shows that increasing dataset size in this way leads to increased empirical privacy. As a complementary study, we investigate scaling the dataset size while maintaining the secret *count*. Concretely, we generate another 200 synthetic author profiles and merge them with TOFU-1. We refer to the obtained dataset as TOFU-2*. Compared to TOFU, TOFU-2* has doubled dataset size but the same secret count; compared to TOFU-2, TOFU-2* has the same dataset size but half secret density. We present the comparison between TOFU, TOFU-2 and TOFU-2* in Fig. 16. TOFU-2* achieves the lowest empirical privacy among all.

**Additional results on consistency of the trends.**

*Secret subset (Fig. 17).* We conduct this experiment using Llama-2-7b and variants of the TOFU dataset. We randomly

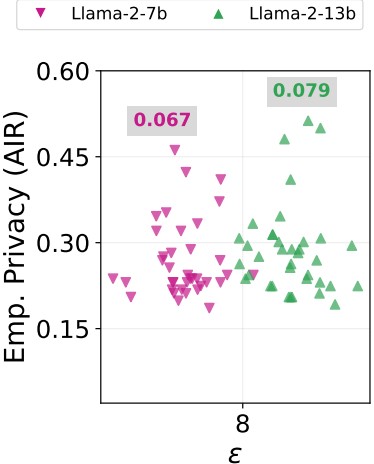

*Figure 15.* **Empirical privacy variance under different model sizes (Llama-2-7b vs. Llama-2-13b).** Each color corresponds to a model size. The figure shows AIR scores achieved by models trained using different configurations. Each group's standard deviation is labeled at the top of its cluster. The results show that Llama-2-13b achieves higher empirical privacy variance than Llama-2-7b.

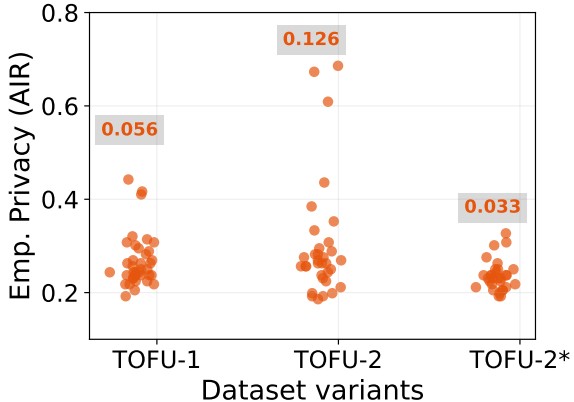

*Figure 16.* **Empirical privacy variance under different data variants (TOFU, TOFU-2, and TOFU-2*)**, at $\varepsilon = 8$. The figure shows AIR scores achieved by models trained using different configurations. Each group's standard deviation is labeled at the top of its cluster. TOFU-2* achieves the lowest empirical privacy variance among the three.

sample half of the secrets (26 out of 52 author-genre pairs) without replacement for three times to create subsets (0, 1, 2). We then measure AIR of these subsets for models trained on different dataset sizes (TOFU-1, TOFU-2, TOFU-4) with varying $\varepsilon$'s. The results show that, across all subsets considered, empirical privacy variance increases as either $\varepsilon$ or dataset size grows.

*Empirical privacy measure (Fig. 18).* We conduct this experiment on the Enron dataset. We measure ACR and VMR for models of different sizes (GPT-2-S and GPT-2-L) trained with varying $\varepsilon$'s. The results show that, for both empirical privacy measures, empirical privacy variance increases as either $\varepsilon$ or model size grows.

## C.2. Does model distance explain the trends of empirical privacy?

One plausible hypothesis for the observed trends in empirical privacy variance is that, a set of models exhibits high empirical privacy variance because the average "distance" (which we will formally define shortly) between them is also large. The intuition behind this hypothesis is fairly straightforward: models that are "close" to each other should have similar empirical privacy scores. This hypothesis can be formally stated as follows: if the average "distance" within $S_1$ is greater than that within $S_2$, then the empirical privacy variance measured on $S_1$ will also be larger than that on $S_2$. Here, $S_1$ and $S_2$ denote

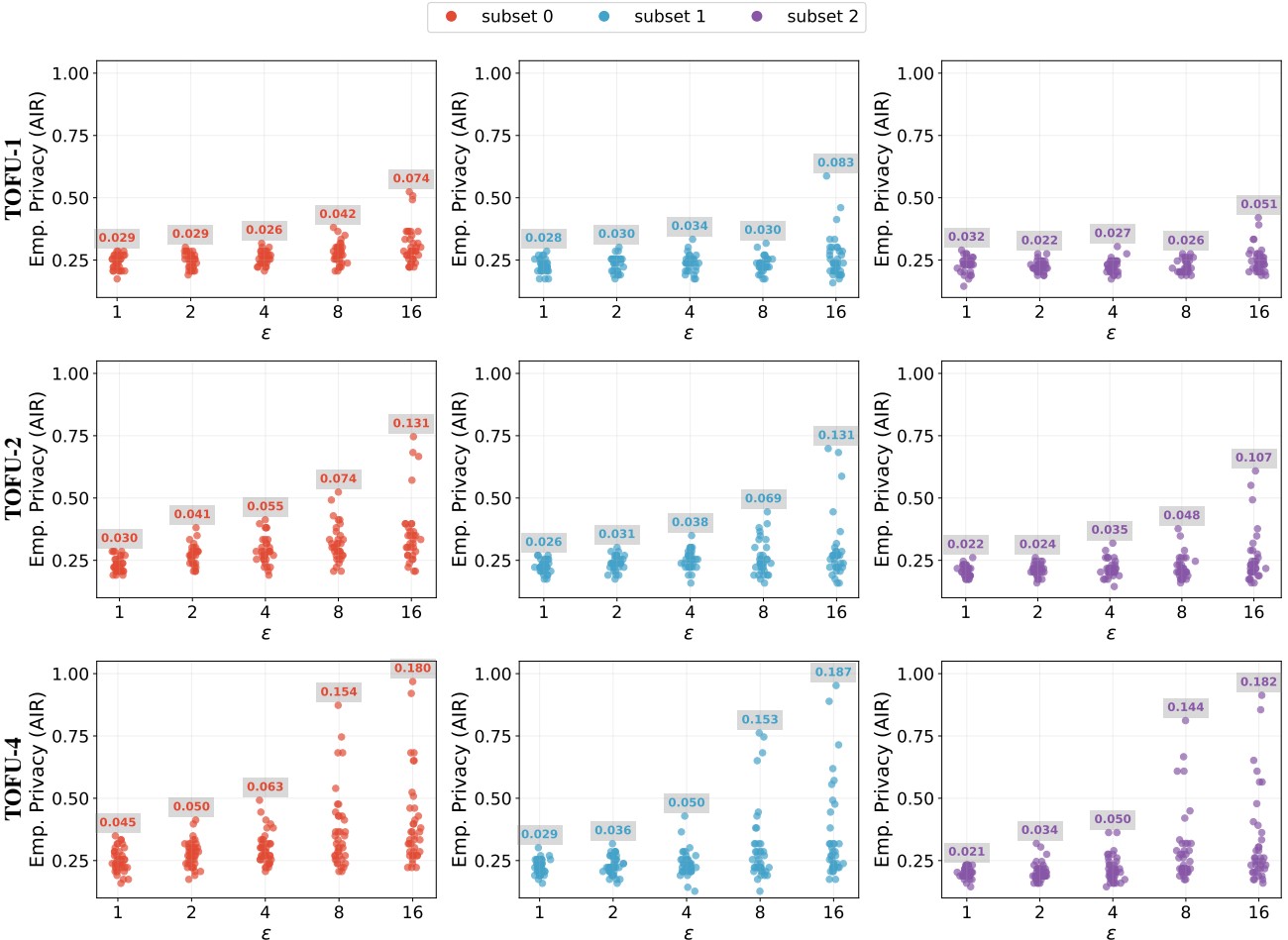

*Figure 17.* **Empirical privacy variance across different secret subsets.** *Columns* correspond to different subsets, and rows correspond to different dataset sizes (TOFU-1, TOFU-2, TOFU-4 from top to bottom). Each subfigure shows AIR values achieved by models trained using different configuration at different $\varepsilon$'s. Each group's standard deviation is labeled at the top of its cluster. The results show that, across all subsets considered, empirical privacy variance increases as either $\varepsilon$ or dataset size grows.

two sets of models, where models within each set share the same architecture and initialization, are trained on the same dataset using DP-SGD, and differ only in their configurations and inherent training randomness.

**Choices of $(S_1, S_2)$ pairs.** We consider three types of $(S_1, S_2)$ pairs corresponding to the three types of trends we identify in Section 3.2: 1) $S_1$ and $S_2$ share model and data, but differ in $\varepsilon$; 2) $S_1$ and $S_2$ share $\varepsilon$ and data, but differ in model size; 3) $S_1$ and $S_2$ share $\varepsilon$ and model, but differ in dataset size.

**Distance metrics.** We compute the average distance over a set of models by the *mean pairwise* distance over all *model pairs*. We consider two distance metrics: 1) parameter space distance—$\ell_2$ distance between model parameters; 2) functional distance—$\ell_2$ distance between model's prediction on a held-out test set, formally:

$$d_f(M_1, M_2) \;=\; \mathbb{E}_{x \sim \mathcal{D}, t \sim [T]}\left[\left\|\mathrm{softmax}\big(M_1(x)_t\big) \;-\; \mathrm{softmax}\big(M_2(x)_t\big)\right\|_2\right], \tag{1}$$

where $M_1$ and $M_2$ are two models, $\mathcal{D}$ stands for the empirical distribution of the held-out test set, $T$ is the number of token positions, and $\mathrm{softmax}(M(x)_t)$ denotes the probability distribution over the vocabulary obtained by applying softmax to the logits.

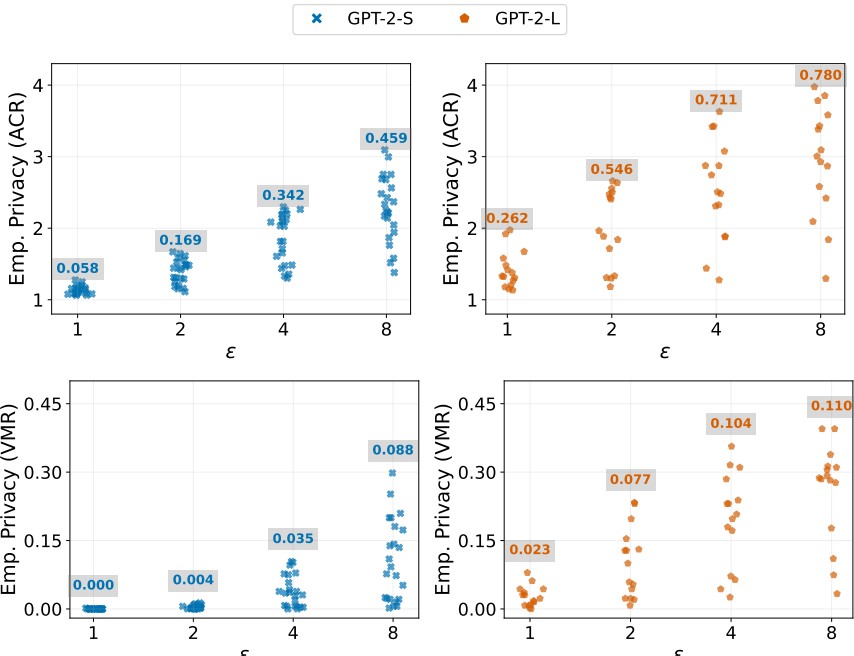

*Figure 18.* **Empirical privacy variance across different empirical privacy measures (ACR and VMR).** *Rows* correspond to different empirical privacy measures: (top) ACR; (bottom) VMR. Columns correspond to different model sizes: (left) GPT-2-S; (right) GPT-2-L. Each subfigure shows empirical privacy scores achieved by models trained using different configuration at different $\varepsilon$'s. Each group's standard deviation is labeled at the top of its cluster. The results show that, for both empirical privacy measures, empirical privacy variance increases as either $\varepsilon$ or model size grows.

**Results.** Table 7 shows that: 1) holding data and model fixed, *larger* $\varepsilon$ has *smaller* average distance; 2) holding data and $\varepsilon$ fixed, *larger* model size has *smaller* average distance; 3) holding model and $\varepsilon$ fixed, varying dataset size does not seem to affect the average distance. These results **refute** the hypothesis and suggest that model distance might not be able to explain the trends of empirical privacy in Section 3.

*Table 7.* **Average distance within model sets with varying $\varepsilon$, model size, or dataset size.** We report functional distance in (b-c).

(a) Vary $\varepsilon$ (model=GPT-2-S, data=Enron)

| $\varepsilon$ | Param. Dist. | Func. Dist. |
|---|---|---|
| 1 | 32.48 | 0.1244 |
| 2 | 32.07 | 0.1143 |
| 4 | 31.74 | 0.1086 |
| 8 | 31.24 | 0.1023 |

(b) Vary model size (data=Enron)

| $\varepsilon$ | GPT-2-S | GPT-2-L |
|---|---|---|
| 1 | 0.1244 | 0.0920 |
| 2 | 0.1143 | 0.0863 |
| 4 | 0.1086 | 0.0824 |
| 8 | 0.1023 | 0.0796 |

(c) Vary dataset size (model=Llama-2-7b)

| $\varepsilon$ | TOFU-1 | TOFU-2 | TOFU-4 |
|---|---|---|---|
| 1 | 0.0580 | 0.0539 | 0.0575 |
| 8 | 0.0637 | 0.0592 | 0.0632 |

### C.3. Impact of hyperparameters vs. impact of random seeds

For all results reported in Section 3.2 and in Appendix C.1, we averaged out the effect of random seeds. Here, we compare the variance induced by random seeds and that induced by hyperparameter configurations. For seeds, we compute the standard deviation of empirical privacy scores across seeds for each configuration and average these values. For configurations, we compute the mean across seeds for each configuration and then the standard deviation of these means. We present the results in Table 8, showing that variance from seeds is approximately *half* that of configurations.

*Table 8.* **Variance induced by inherent randomness in model training vs. variance induced by hyperparameters.** The numbers reported in the table are standard deviations.

| $\varepsilon$ | GPT-2-S, Enron, ACR | | GPT-2-L, Enron, ACR | |
| --- | --- | --- | --- | --- |
| | randomness | hyperparameter | randomness | hyperparameter |
| 1 | 0.06 | 0.06 | 0.14 | 0.26 |
| 2 | 0.11 | 0.16 | 0.26 | 0.48 |
| 4 | 0.16 | 0.32 | 0.28 | 0.56 |
| 8 | 0.19 | 0.41 | 0.31 | 0.58 |

# D. Additional Experimental Results for Section 4

## D.1. Additional results on accuracy of heuristics

Fig. 19 shows the complete set of results corresponding to all combinations of models, datasets, and empirical privacy measures, evaluated at varying $\varepsilon$ values.

The results show that our heuristics outperform the random guess baseline. Notably, on the TOFU datasets, heuristic accuracy improves with increasing $\varepsilon$ and larger dataset sizes (see Fig. 19(3a–3e)).

For the specific setting of (GPT-2-S, Enron, VMR, $\varepsilon = 1$), the accuracy of all three heuristics is close to 0. This is due to an artifact where the VMR scores are nearly all 0 for all configurations at $\varepsilon = 1$, meaning no configuration is distinguishably better than another, leading to the observed low accuracy.

More evaluation results on different density groups of *density-adjusted* TOFU can be found in Fig. 35 in Appendix H.

## D.2. Visualization of selection quality

Fig. 20(a) illustrates the layout of empirical privacy and utility for all configurations, serving as the basis of our selection process. We progressively slide a utility threshold $u$ from left to right (high to low utility), and at each threshold, each selection method chooses a configuration from the corresponding subpool $P_u$.

**Results.** Fig. 20(b) presents the empirical privacy scores (AIR) of configurations selected by different methods across all thresholds. The visualization offers a more fine-grained and intuitive comparison of the selection quality of different methods. As a reference, the **oracle** points collectively form the Pareto front. We observe that the **best-utility practice** prioritizes utility at the cost of empirical privacy, while the **worst-utility heuristic** appears unstable and overly sensitive to individual points. In contrast, **our procedure** exhibits near-oracle behavior, ensuring stable and robust performance across all threshold levels. A full set of demonstration results can be found at Figs. 32 to 34 in Appendix H.

## D.3. Additional results on relative privacy risk

We evaluate all combinations of **models**, **datasets**, **empirical privacy measures**, and $\varepsilon$ **values**. Each combination corresponds to a specific layout of models for the hyperparameter selection task (see Fig. 20(a)). Before discussing the results, we introduce additional considerations in the experimental setup.

**Accounting for training randomness.** In Fig. 20(a), each point represents an average over multiple random seeds, meaning the observed layout is just one realization drawn from an underlying distribution. This simplification averages out the impact of training randomness.

To better account for the variation in privacy risks, we sample layouts using a Monte Carlo approach. Specifically, we model each configuration as a Gaussian distribution, with its mean and standard deviation estimated from empirical data (i.e., across random seeds). This allows us to generate multiple plausible layouts and analyze the effectiveness and robustness of our selection method under training randomness. In our experiments, we adopt the number of trials as 5,000.

**A complementary metric: absolute privacy risk** The relative privacy risk metric introduced in the main paper runs into issues when the oracle's privacy risk is zero, which occurs in the case of VMR. Since relative comparisons become ill-defined in such cases, we also compute an absolute privacy risk measure to ensure meaningful evaluations across all

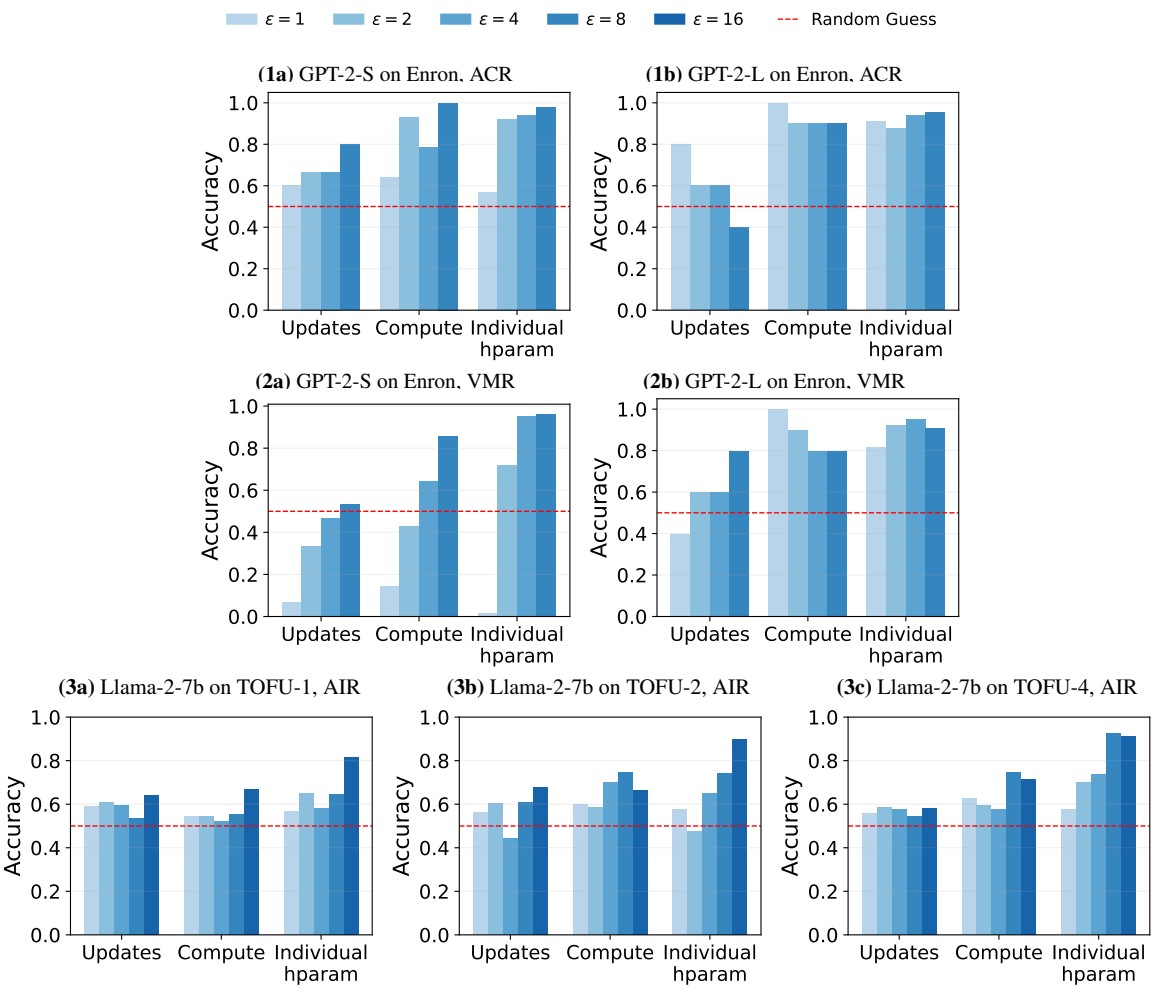

*Figure 19.* Accuracy of the three heuristics across different **models**, **datasets**, and **empirical privacy measures**, evaluated at varying $\varepsilon$'s.

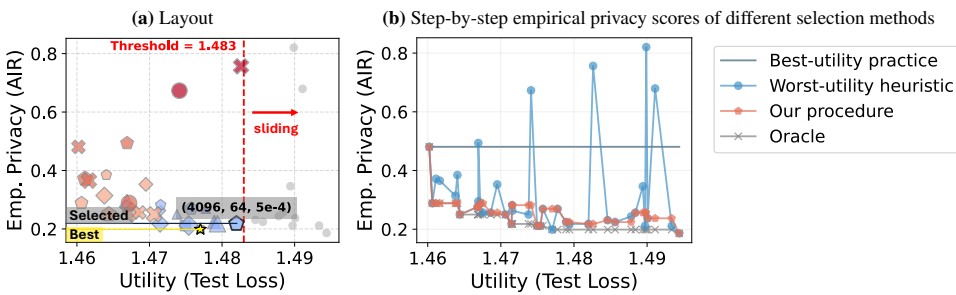

*Figure 20.* Setting=(Llama-2-7b, TOFU-4, $\varepsilon = 8$). **(a)** The layout of empirical privacy (AIR) vs. utility (test loss) achieved by all configurations calibrated to the same DP guarantee. **(b)** At each utility threshold $u$, each of the considered method (oracle, best-utility practice, worst-utility heuristic, and our procedure) will make a selection from the subpool $P_u$. We plot the scores of the selected points against the thresholds.

settings.

**Results.** The results are presented in Figs. 21 to 23. Our procedure outperforms the baselines in almost all settings, demonstrating the effectiveness of the underlying heuristics and their ability to generalize. More evaluation results on different density groups of *density-adjusted* TOFU can be found in Fig. 36 in Appendix H.

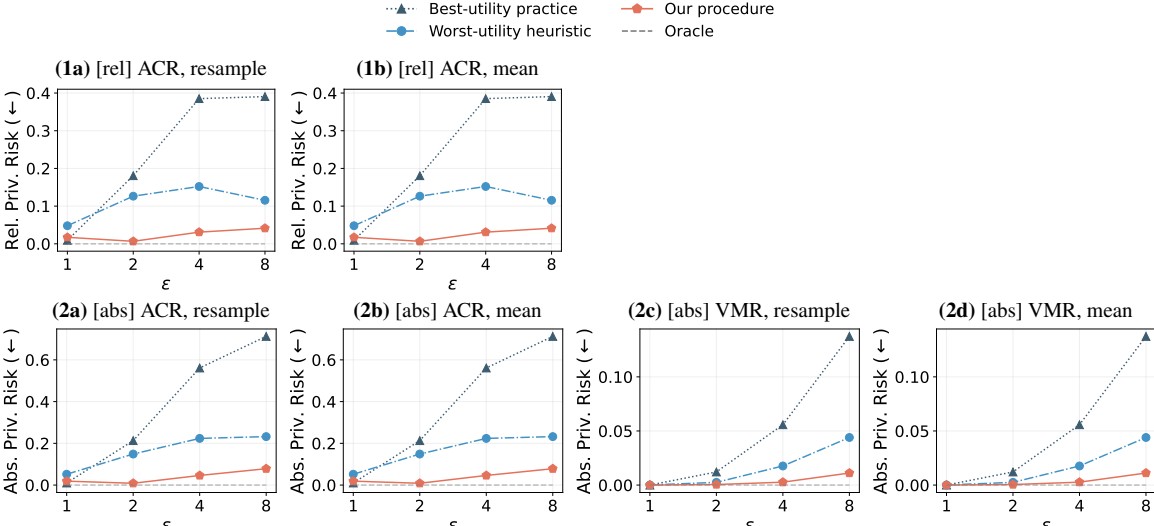

*Figure 21.* **Relative privacy risk of our procedure compared with baselines.** Model = GPT-2-S; data = Enron; empirical privacy measure $\in$ {ACR, VMR}; risk measure $\in$ {abs, rel}, where "abs" denotes the absolute privacy risk and "rel" denotes for the relative privacy risk; layout $\in$ {resample, mean}, where "resample" corresponds to the monte carlo approach that accounts for the training randomness, and "mean" corresponds to the approach that directly uses the mean, averaging out the training randomness. *Note that GPT-2-S achieves VMR of 0 at multiple configurations, which makes the relative risk metric invalid. Thus we omit the relative risk results for VMR and present results for the absolute risk only.*

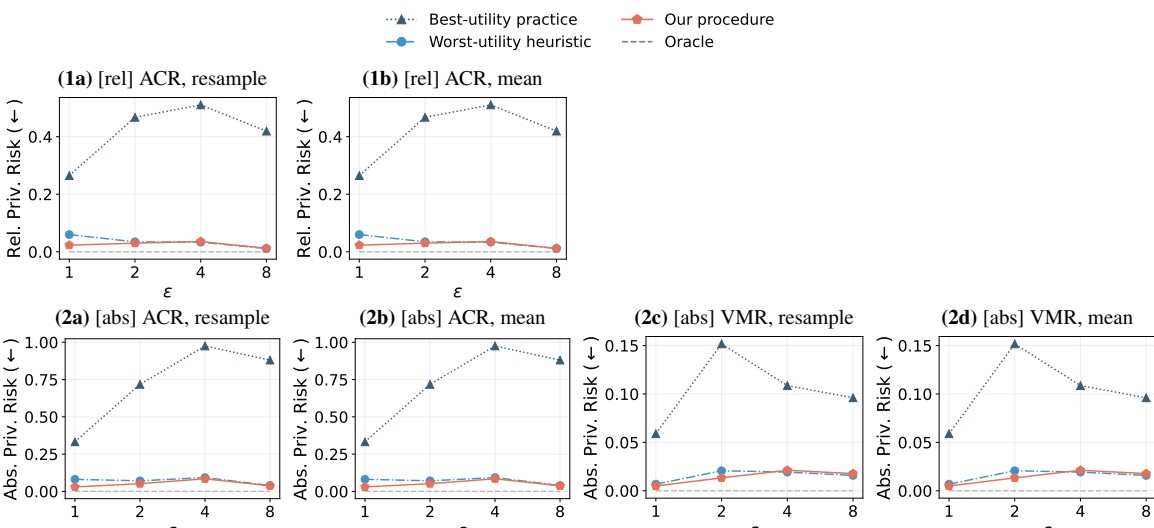

*Figure 22.* **Relative privacy risk of our procedure compared with baselines.** Model = GPT-2-L, data = Enron, empirical privacy measure $\in$ {ACR, VMR}; risk measure $\in$ {abs, rel}; layout $\in$ {resample, mean}. *Note that GPT-2-L achieves VMR of 0 at multiple configurations, which makes the relative risk metric invalid. Thus we omit the relative risk results for VMR and present results for the absolute risk only.*

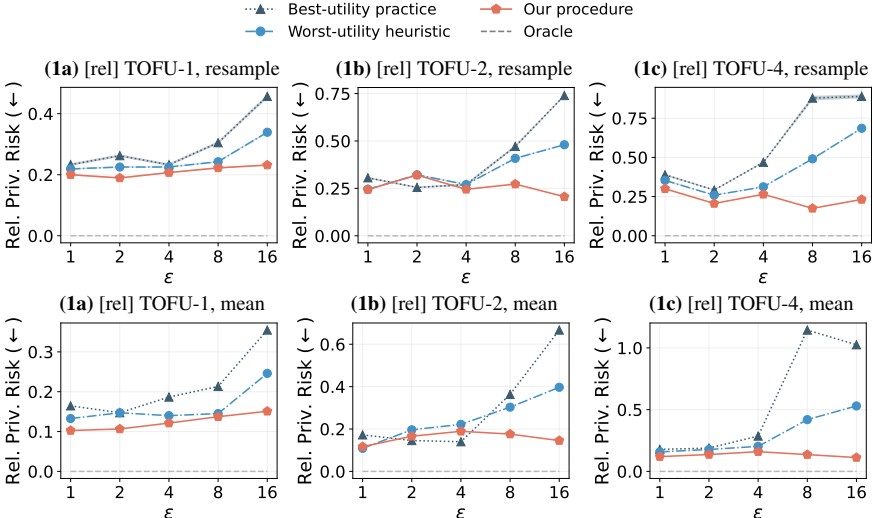

*Figure 23.* **Relative privacy risk of our procedure compared with baselines.** Model = Llama-2-7b, data $\in$ {TOFU-1, TOFU-2, TOFU-4}, measure = AIR, risk = rel, layout = resample.

# E. Additional Experimental Setup and Results for Section 5

## E.1. Privacy auditing setup and results

### E.1.1. EXPERIMENTAL SETUPS

We follow the approach of Panda et al. (2025) to perform privacy auditing of fine-tuned LLMs. Our experiments focus on a single setting: fine-tuning GPT-2-S on the Enron dataset with a privacy budget of $\varepsilon = 4$ and $\varepsilon = 8$. We experiment with both full fine-tuning and LoRA fine-tuning and find that the method mostly gives meaningful $\hat{\varepsilon}$ for full fine-tuning (i.e., $\hat{\varepsilon} > 1$) but not always for LoRA fine-tuning. Thus we stick to full fine-tuning in this study.

**High-level procedure.**     As outlined in Section 5.1, we aim to address two questions: Q1) *Does $\hat{\varepsilon}$ vary with configurations?* Q2) *Is $\hat{\varepsilon}$ aligned with empirical privacy?* To answer these, we follow a structured approach.

1. Define a range of hyperparameter configurations (as shown in Table 9).

2. For each configuration,

    (a) Perform DP fine-tuning for each configuration on the Enron dataset with input canaries injected (explained below).
    (b) After fine-tuning, i) measure the model's $\hat{\varepsilon}$ using the auditing procedure and ii) calculate its ACR.

3. Finally, analyze the measured $\hat{\varepsilon}$'s and empirical privacy scores on all the obtained models: 1) examine how $\hat{\varepsilon}$ varies with different configurations; 2) assess the correlation between $\hat{\varepsilon}$ and the empirical privacy score.

*Table 9.* **Hyperparameter configurations for full fine-tuning GPT-2 models on Enron with input canaries.** The tuple represents $(b, T, \eta, c)$. For each configuration, we perform fine-tuning using 4 random seeds.

| Scenario | Configurations | | |
|---|---|---|---|
| **GPT-2-S, Enron** (total=7) $\varepsilon = 4, 8$ | $(8192, 1000, 1 \times 10^{-3}, 0.5)$ $(8192, 250, 5 \times 10^{-4}, 0.5)$ $(4096, 250, 1 \times 10^{-3}, 0.5)$ $(2048, 250, 1 \times 10^{-3}, 0.5)$ | $(8192, 250, 1 \times 10^{-3}, 0.5)$ $(8192, 250, 3 \times 10^{-3}, 0.5)$ | $(8192, 125, 1 \times 10^{-3}, 0.5)$ |

**Details of privacy auditing.**     Privacy auditing quantifies a model's privacy leakage by measuring the ability to distinguish member and non-member samples. The auditing procedure compares the losses of member and non-member canaries, then converts these loss differences into $\hat{\varepsilon}$ using the method proposed by Steinke et al. (2024).

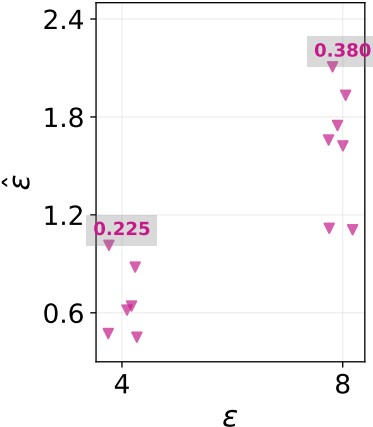

*Figure 24.* **Variance of $\hat{\varepsilon}$.** We follow Panda et al. (2025) to produce an audited lower bound, i.e., $\hat{\varepsilon}$, for each fine-tuned LLM. Each scattered point corresponds to one configuration.

Panda et al. (2025) introduce "new token canaries", which improve canary design to better expose memorization. To further maximize exposure, we adopt the following design choices, as recommended by the authors: 1) Initialize the token embeddings for new tokens to zero; 2) Precede each canary sample with a unique random sequence; 3) Compute the SFT loss only on the last new token for each canary sample. These design enhancements ensure that the input canaries are highly effective in detecting privacy leakage.

### E.1.2. EXPERIMENTAL RESULTS

**A1: $\hat{\varepsilon}$ varies with configurations.** Fig. 24 shows that, for both $\varepsilon$ values we consider ($\varepsilon = 4$ and $\varepsilon = 8$), different configurations could lead to different $\hat{\varepsilon}$'s.

**A2: $\hat{\varepsilon}$ is not aligned with empirical privacy.** Given the models trained under differnet configurations, we assess the correlation between their $\hat{\varepsilon}$ and the empirical privacy scores (ACR). For $\varepsilon = 8$, the spearman correlation between $\hat{\varepsilon}$ and ACR is -0.13.

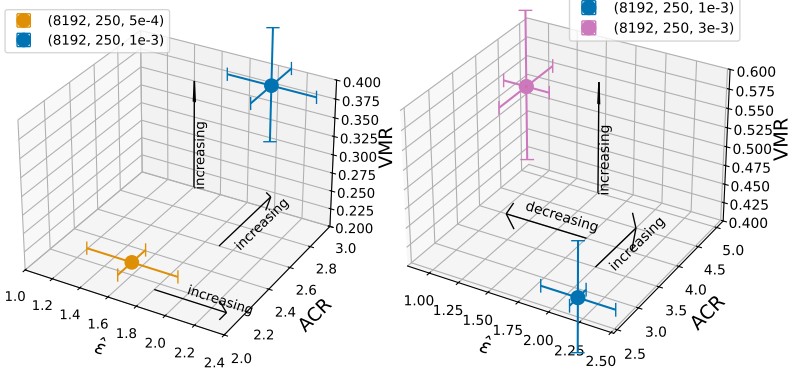

*Figure 25.* Effect of learning rate $\eta$ on $\hat{\varepsilon}$ and empirical privacy measures (ACR and VMR). As $\eta$ increases, empirical privacy measures increase, but $\hat{\varepsilon}$ decreases, highlighting a misalignment.

*Case study on learning rate $\eta$.* Fig. 25 visualizes how $\hat{\varepsilon}$ and empirical privacy scores (ACR, VMR) change as $\eta$ increases from $5 \times 10^{-4}$ to $1 \times 10^{-3}$ further to $3 \times 10^{-3}$. While ACR and VMR both keep increasing, $\hat{\varepsilon}$ starts to drop in the second half (as $\eta$ increases from $1 \times 10^{-3}$ to $3 \times 10^{-3}$).

*Role of utility.* We additionally note in the above example that the change of $\hat{\varepsilon}$ closely follows the change in utility (which first improves from 2.94 to 2.87, and then degrades to 3.08 at the largest $\eta$). We thus introduce the quantity utility into the

picture and measure the Spearman rank correlation between $\hat{\varepsilon}$, ACR and utility, showing a -0.71 correlation for $(\hat{\varepsilon}, \text{utility})$ and a -0.05 correlation for $(\text{ACR}, \text{utility})$. This indicates that $\hat{\varepsilon}$ and empirical privacy are not aligned, but instead $\hat{\varepsilon}$ is strongly correlated with utility.

### E.2. Additional results on privacy profile

**Experimental setup and results.** We extend our privacy profile analysis in Section 5.2. Specifically, we first fix the dataset size at $n = 180,000$ and set the target privacy requirement to $(4, 10^{-6})$-DP, then consider:

*Scenario 1:* Fix $b$, vary $T$: we analyze the effect of increasing $T \in \{150, 500, 2000\}$ while keeping $b = 4096$ fixed.

*Scenario 2:* Fix $T$, vary $b$: we analyze the effect of increasing $b \in \{2048, 4096, 8192\}$ while keeping $T = 1000$ fixed.

We also consider the following privacy accountants: Rényi Differential Privacy (RDP) accountant (Mironov, 2017), PRV accountant (Gopi et al., 2021), and Connect-the-Dots accountant (Doroshenko et al., 2022).

Table 10 summarizes the noise multipliers $\sigma$ required to achieve $(4, 10^{-6})$-DP in the two scenarios, as computed by the three privacy accountants. We observe that RDP provides a looser bound, while PRV and Connect-the-Dots yield similarly tight estimates (as reflected by their noise multipliers).

*Table 10.* The noise multiplier returned by different privacy accountants in two scenarios. **Setup 1**: $n = 180,000$, target=$(4, 10^{-6})$-DP

| Scenario | | RDP | PRV | Connect-the-Dots |
|---|---|---|---|---|
| **1. Fix $b$ vary $T$** $n = 180,000, b = 4096$ | $T = 150$ | $\sigma = 8.52 \times 10^{-1}$ | $\sigma = 8.00 \times 10^{-1}$ | $\sigma = 7.99 \times 10^{-1}$ |
| | $T = 500$ | $\sigma = 1.01$ | $\sigma = 9.62 \times 10^{-1}$ | $\sigma = 9.61 \times 10^{-1}$ |
| | $T = 2000$ | $\sigma = 1.50$ | $\sigma = 1.43$ | $\sigma = 1.43$ |
| **2. Fix $T$ vary $b$** $n = 180,000, T = 1000$ | $b = 2048$ | $\sigma = 8.57 \times 10^{-1}$ | $\sigma = 8.14 \times 10^{-1}$ | $\sigma = 8.14 \times 10^{-1}$ |
| | $b = 4096$ | $\sigma = 1.20$ | $\sigma = 1.14$ | $\sigma = 1.14$ |
| | $b = 8192$ | $\sigma = 2.01$ | $\sigma = 1.91$ | $\sigma = 1.91$ |

Fig. 26 presents the corresponding privacy profiles, revealing the following *consistent* patterns:

- The privacy profiles intersect at only one point, corresponding to the target privacy requirement.

- For a fixed target privacy requirement, increasing $\sigma$ (which corresponds to a larger $T$ or $b$) decreases $\varepsilon$ in the bottom-right (small $\delta$ regime) but increases $\varepsilon$ in the top-left (large $\delta$ regime).

We further study the impact of $\varepsilon$ and dataset size. We vary $\varepsilon \in \{1, 2, 4, 8\}$ under *Scenario 1* and present the results in Fig. 27, and vary $n \in \{45k, 90k, 180k, 360k\}$ under *Scenario 1* and present the results in Fig. 28. We show that the difference between privacy profiles at different $T$'s enlarge as $\varepsilon$ increases (or as $n$ increases), when the noise multiplier $\sigma$ decreases (see Table 12 for the calculated noise multiplier $\sigma$ under different settings).

**An alternative setup and results.** We also consider an alternative experimental setup with $n = 50,000$ and a target privacy requirement of $(8, 5 \times 10^{-6})$-DP. In scenario 1, we fix $b = 4096$ and vary $T \in \{250, 500, 1000\}$. In scenario 2, we fix $T = 500$ and vary $b \in \{1024, 2048, 4096\}$. The results are presented in Table 11 and Fig. 29. Despite the variations in setup, the patterns of privacy profiles remain consistent.

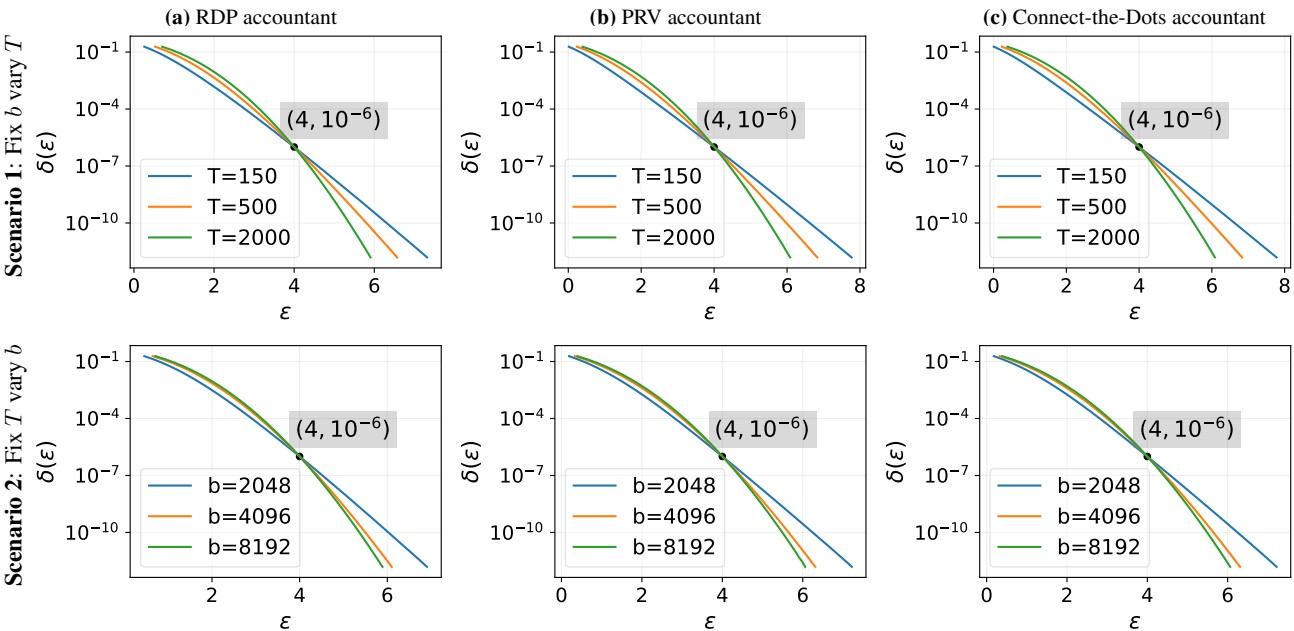

*Figure 26.* **Privacy profiles under different privacy accountants, setup 1**: $n = 180,000$, target=$(4, 10^{-6})$-DP. Columns correspond to different privacy accountants: (a) RDP accountant, (b) PRV accountant, and (c) Connect-the-Dots accountant. The first row (Scenario 1) shows the impact of varying $T$ while fixing $b = 4096$, and the second row (Scenario 2) shows the effect of varying $b$ while fixing $T = 1000$.

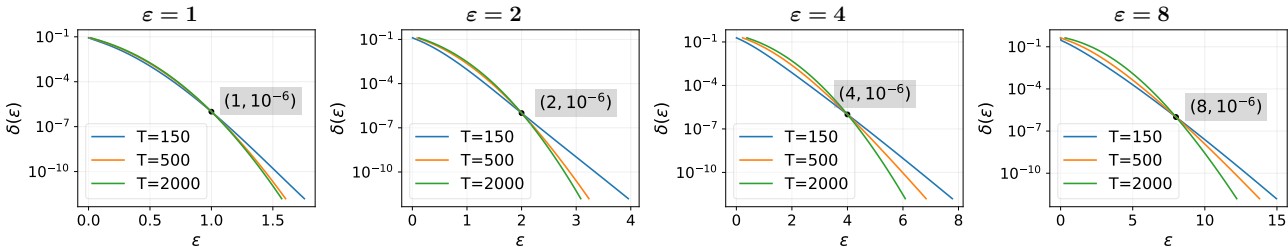

*Figure 27.* **Privacy profiles for different target $\varepsilon \in \{1, 2, 4, 8\}$ (different columns).** $n = 180,000$, $b = 4096$, target = $(\varepsilon, 10^{-6})$-DP, accountant = Connect-the-Dots accountant. The difference between the privacy profiles of different $T$'s enlarge as $\varepsilon$ increases.

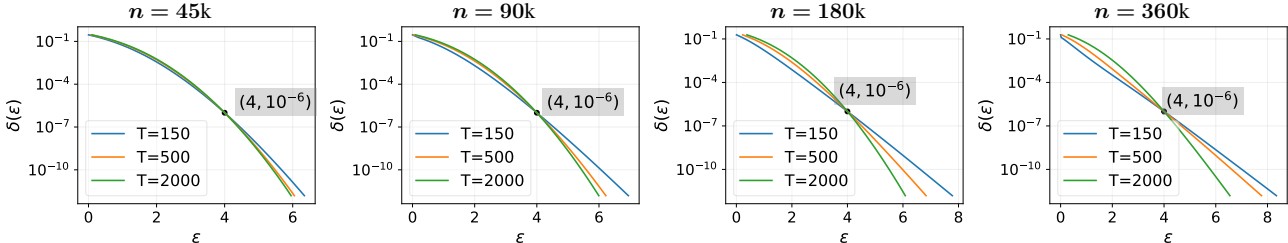

*Figure 28.* **Privacy profiles for different dataset size $n \in \{45k, 90k, 180k, 360k\}$ (different columns).** $b = 4096$, target = $(4, 10^{-6})$-DP, accountant = Connect-the-Dots accountant. The difference between the privacy profiles of different $T$'s enlarge as $n$ increases.

# F. Results under the Worst-Case Privacy Measure

In this section, we investigate whether switching to a worst-case empirical privacy measure has any impact on the conclusions we obtained in the main paper. We achieve this by taking the max of the measured empirical privacy scores on the set of

*Table 11.* The noise multiplier returned by different privacy accountants in two scenarios. **Setup 2**: $n = 50,000$, target=$(8, 5 \times 10^{-6})$-DP

| Scenario | | RDP | PRV | Connect-the-Dots |
|---|---|---|---|---|
| **1. Fix $b$ vary $T$**
$n = 50,000, b = 4096$ | $T = 250$ | $\sigma = 1.15$ | $\sigma = 1.09$ | $\sigma = 1.09$ |
| | $T = 500$ | $\sigma = 1.43$ | $\sigma = 1.36$ | $\sigma = 1.36$ |
| | $T = 1000$ | $\sigma = 1.87$ | $\sigma = 1.77$ | $\sigma = 1.77$ |
| **2. Fix $T$ vary $b$**
$n = 50,000, T = 500$ | $b = 1024$ | $\sigma = 7.06 \times 10^{-1}$ | $\sigma = 6.72 \times 10^{-1}$ | $\sigma = 6.72 \times 10^{-1}$ |
| | $b = 2048$ | $\sigma = 9.33 \times 10^{-1}$ | $\sigma = 8.90 \times 10^{-1}$ | $\sigma = 8.89 \times 10^{-1}$ |
| | $b = 4096$ | $\sigma = 1.43$ | $\sigma = 1.36$ | $\sigma = 1.36$ |

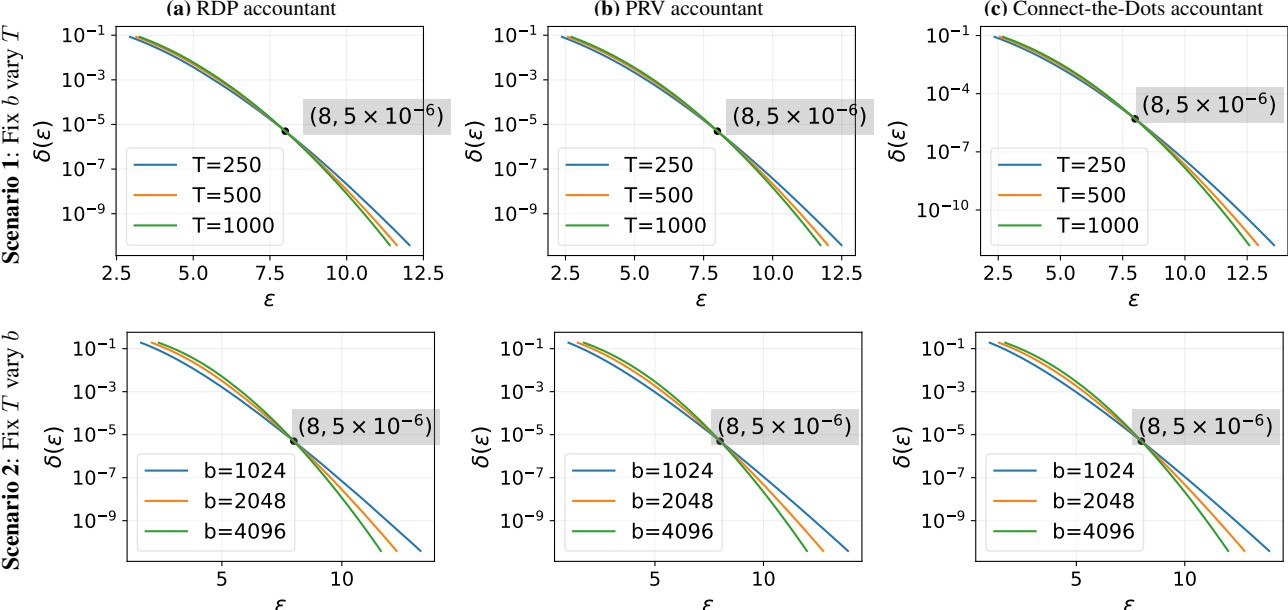

*Figure 29.* **Privacy profiles under different privacy accountants, setup 2**: $n = 50,000$, target=$(8, 5 \times 10^{-6})$-DP. Columns correspond to different privacy accountants: (a) RDP accountant, (b) PRV accountant, and (c) Connect-the-Dots accountant. The first row (Scenario 1) shows the impact of varying $T$ while fixing $b = 4096$, and the second row (Scenario 2) shows the effect of varying $b$ while fixing $T = 500$.

secrets, instead of the average as done in the main paper (see the end of Section 3.1).

We present detailed results below, supporting that all conclusions remain unchanged.

**Empirical privacy.** Fig. 30 shows that empirical privacy variance happens for both the average-case and the worst-case privacy measure (the former is the same as Fig. 2(a)).

**Regression results.** In Table 13, we present the regression results using the *worst-case* privacy measure as the response variable $y$, and compare with that obtained using the *average-case* privacy measure (as presented in Table 2 in the main paper).

As shown in Table 13, the conclusions drawn on the *average-case* regression results (in Section 4.1) remain to hold on the *worst-case* regression results—(1) all individual hyperparameters have positive coefficients with significant $p$-values, thus increasing any individual hyperparameter leads to worse empirical privacy; (2) The coefficient of $\log b$ is smallest, meaning that under fixed compute, increasing $b$ (while decreasing $T$ proportionally) improves empirical privacy; (3) The coefficient of $\log \eta$ is larger than $\log C$, meaning under fixed updates, decreasing $\eta$ (while increasing $C$ proportionally) improves empirical privacy.

**Correlation with audited $\hat{\varepsilon}$.** We measure the spearman rank correlation between the audited $\hat{\varepsilon}$ and the worst-case empirical privacy score, replicating the exercise in Section 5.1. Similar to the conclusion obtained on the average-case privacy measure

*Table 12.* **The noise multiplier $\sigma$ achieved at different settings.** For all settings, we fix $b = 4096$ and set the target to $(4, 10^{-6})$-DP; accountant = Connect-the-Dots. (Left): vary $\varepsilon \in \{1, 2, 4, 8\}$ and fix $n = 180k$; (right): vary $n \in \{45k, 90k, 180k, 360k\}$ and fix $\varepsilon = 4$.

| Varying $\varepsilon$ | 1 | 2 | 4 | 8 |
|---|---|---|---|---|
| $T = 150$ | 1.51 | 1.05 | 0.80 | 0.62 |
| $T = 500$ | 2.34 | 1.41 | 0.96 | 0.72 |
| $T = 2000$ | 4.40 | 2.42 | 1.43 | 0.96 |

| Varying $n$ | 45k | 90k | 180k | 360k |
|---|---|---|---|---|
| $T = 150$ | 1.61 | 1.05 | 0.80 | 0.67 |
| $T = 500$ | 2.60 | 1.46 | 0.96 | 0.75 |
| $T = 2000$ | 4.94 | 2.56 | 1.43 | 0.92 |

*Figure 30.* **Empirical privacy variance happens for both average-case (left) and worst-case (right) privacy measures.**

*Table 13.* **Comparison on regression results on the average-case privacy measure vs. the worst-case privacy measure.**

(a) Regression on *individual* hyperparameters

| Variable | Enron *(average-case)* | | Enron *(worst-case)* | |
|---|---|---|---|---|
| | Coef. | $p$-value | Coef. | $p$-value |
| Batch size ($\log b$) | 0.13*** | $1 \times 10^{-5}$ | 0.38** | $2 \times 10^{-3}$ |
| Iterations ($\log T$) | 0.37*** | $< 2 \times 10^{-16}$ | 1.31*** | $5 \times 10^{-14}$ |
| Learning rate ($\log \eta$) | 0.51*** | $5 \times 10^{-15}$ | 1.80*** | $9 \times 10^{-12}$ |

(b) Regression on *composite* hyperparameters

| Variable | Enron *(average-case)* | | Enron *(worst-case)* | |
|---|---|---|---|---|
| | Coef. | $p$-value | Coef. | $p$-value |
| Compute ($\log C$) | 0.22*** | $2 \times 10^{-12}$ | 0.74*** | $3 \times 10^{-9}$ |
| Learning rate ($\log \eta$) | 0.53*** | $6 \times 10^{-13}$ | 1.89*** | $2 \times 10^{-10}$ |

*Notes:* ***$p < 0.001$, **$p < 0.01$, *$p < 0.05$. The response variable (empirical privacy score $y$) is ACR.

in Section 5.1, here we obtain a correlation of -0.29 between the two, showing that the two are not positively correlated. Moreover, the correlation between the average-case privacy measure and the worst-case privacy measure is as high as 0.90.

# G. Compute Resources and Runtime

**Compute resources.** We run our experiments on three main computing environments. The first setup has four NVIDIA H100 GPUs (each with 80GB of HBM3 memory) and an Intel Xeon Platinum 8468 CPU (192 cores). The second setup also has four NVIDIA A100 GPUs (each with 80GB) and an AMD EPYC 7643 CPU (192 cores). For larger-scale experiments, we use a cluster consisting of 100 nodes, where each node contains four 40GB A100 GPUs. We manage all jobs on the cluster using the SLURM scheduling system.

Empirically, training on an H100 achieves an approximately *two-fold speedup* over an A100. To provide a clear sense of computational requirements, below we report GPU hours in H100 unit.

**Runtime.** Runtime consists of both training and evaluation phases.

*Training runtime.* We conduct all fine-tuning experiments on NVIDIA H100 or NVIDIA A100 GPUs, using a single GPU per run. Empirically, training on an H100 achieves an approximately *two-fold speedup* over an A100. To provide a clear sense of computational requirements, we report GPU hours measured on H100 for both the smallest and largest compute configurations, as runtime scales approximately *linearly* with compute $C$.

*Table 14.* Fine-tuning runtime for different compute configurations on different (model, dataset) pairs.

| Model | Configuration $(b, T)$ | Compute $C$ | Runtime (H100) |
|---|---|---|---|
| *GPT-2 Models on Enron* | | | |
| GPT-2-S (smallest) | (1024, 125) | 128,000 | 7.5 min |
| GPT-2-S (largest) | (8192, 1000) | 8,192,000 | 7.5 hrs |
| GPT-2-L (smallest) | (1024, 125) | 128,000 | 30 min |
| GPT-2-L (largest) | (8192, 500) | 4,096,000 | 17 hrs |
| *Llama-2 Models on TOFU* | | | |
| Llama-2-7B (largest) | (4096, 256) | 1,048,576 | 5 hrs |
| Llama-2-13B (largest) | (4096, 256) | 1,048,576 | 8.5 hrs |

- *GPT-2 models on Enron*: For GPT-2-S, the smallest compute configuration, $(b, T) = (1024, 250)$ with $C = 256,000$, completes in 15 minutes, while the largest configuration, $(8192, 1000)$ with $C = 8,192,000$, requires approximately 7.5 hours. For GPT-2-L, the smallest setting, $(1024, 125)$ with $C = 128,000$, has a training time of 30 minutes, whereas the largest configuration, $(8192, 500)$ with $C = 4,096,000$, takes 17 hours.

- *Llama-2 models on TOFU*. For Llama-2 models, the most computationally expensive configuration, $(b, T) = (4096, 256)$ with $C = 1,048,576$, completes in 5 hours for Llama-2-7B and 8.5 hours for Llama-2-13B. Runtime scales proportionally for smaller configurations.

A summary of the above is provided in Table 14. In total, our fine-tuning experiments across various models, datasets, and configurations accumulate over 5,000 GPU hours (H100-equivalent compute).

*Evaluation runtime.* For evaluation, VMR and AIR involve only prompting and is therefore lightweight. In contrast, ACR evaluation is significantly more expensive due to the nature of the search algorithm.

The runtime of ACR per secret per model varies considerably—the search algorithm finishes quickly for a well-memorized secret, but can keep going on for long for a secret that is barely memorized. We present a histogram of the runtime for individual search runs (per model per secret) in Fig. 31; note that both $x$ and $y$ axes are on log scale. On average, ACR evaluation on *one* single secret takes 1,100 seconds for GPT-2-S and 5,300 seconds for GPT-2-L. The total compute for evaluation amounts to approximately 5,000 GPU hours (H100-equivalent compute).

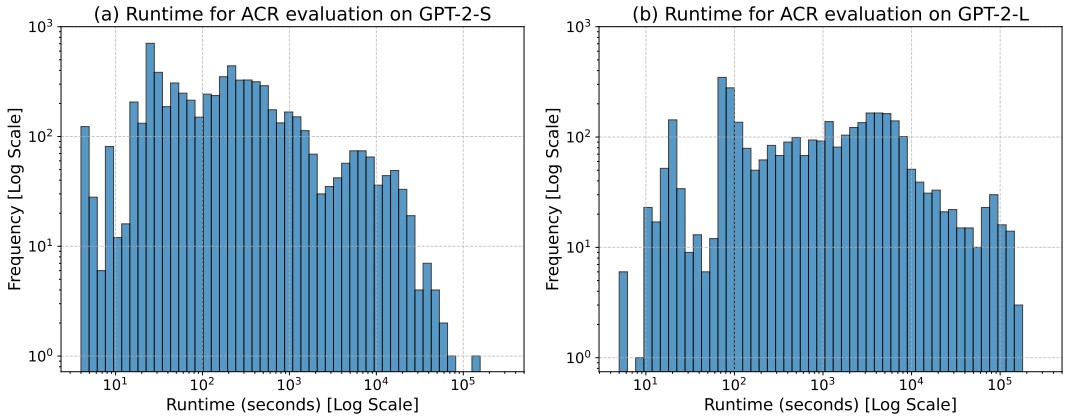

*Figure 31.* **Distribution of evaluation runtime for ACR** for (a) GPT-2-S and (b) GPT-2-L. The $x$-axis represents runtime in seconds on a logarithmic scale, while the $y$-axis indicates the frequency of occurrences (also on log scale).

# H. Complete Sets of Results

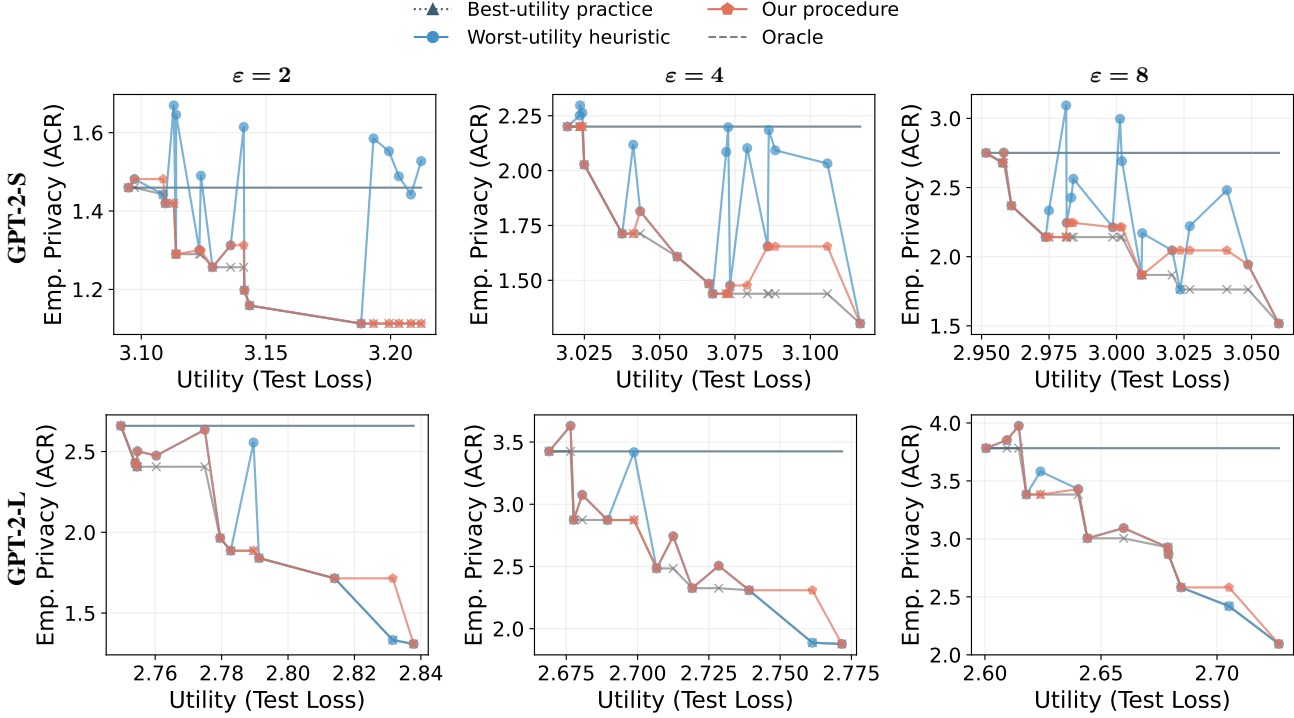

*Figure 32.* **A complete set of visualizations on the selection quality of different methods. Dataset: Enron; Model: GPT-2 models.** **our procedure** *consistently outperforms* the others (i.e., **best-utility practice** and **worst-utility heuristic**) for both models (GPT-2-S and GPT-2-L) and across different $\varepsilon$ values.

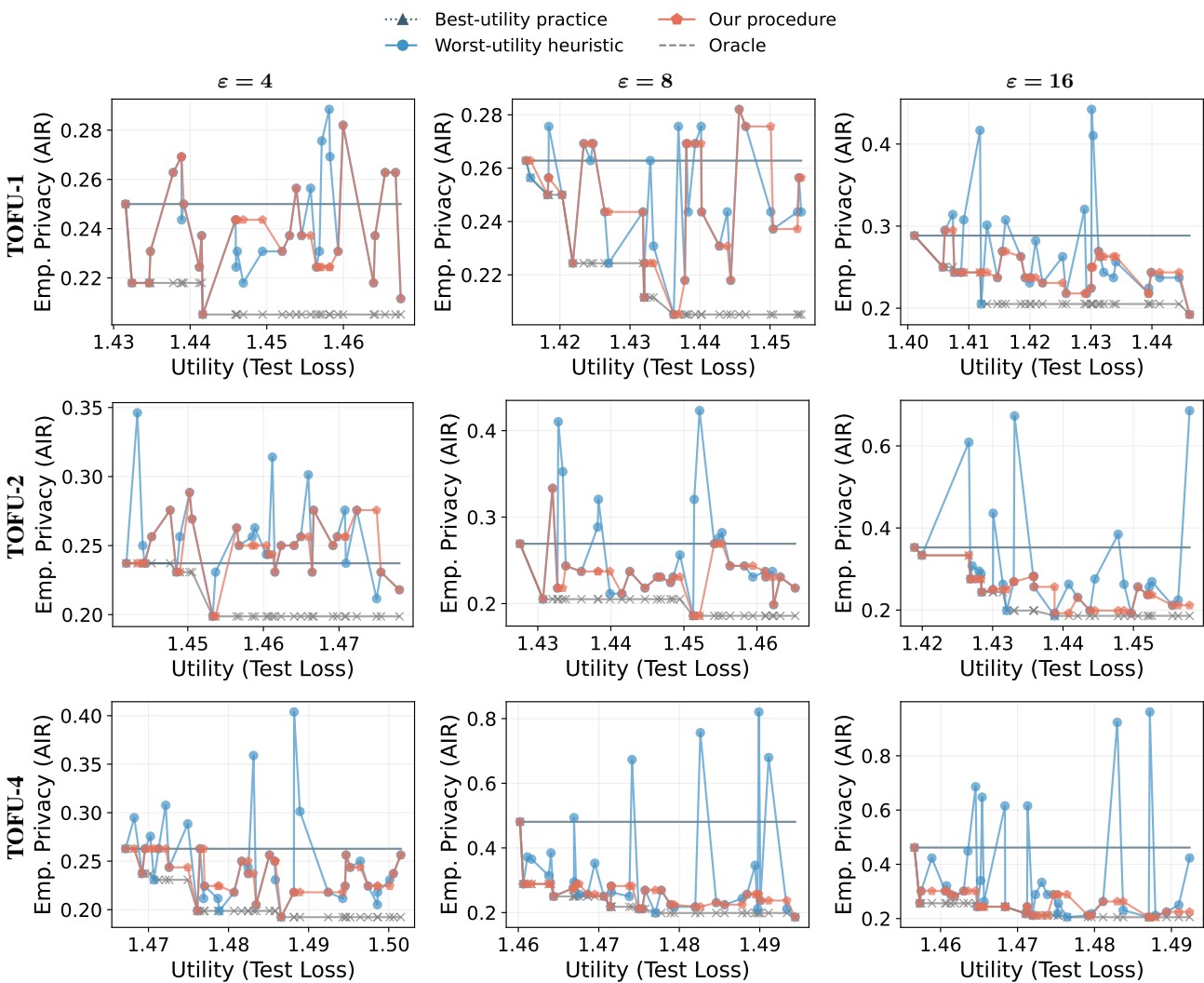

*Figure 33.* **A complete set of visualizations on the selection quality of different methods. Dataset:** *paraphrase-scaled* **TOFU; Model: Llama-2-7b.** The advantage of **our procedure** over the others (i.e., **best-utility practice** and **worst-utility heuristic**) enlarges with the increase of the dataset size (vertically, from TOFU-1 to TOFU-4) and the increase of $\varepsilon$ (horizontally, from $\varepsilon = 4$ to $\varepsilon = 8$).

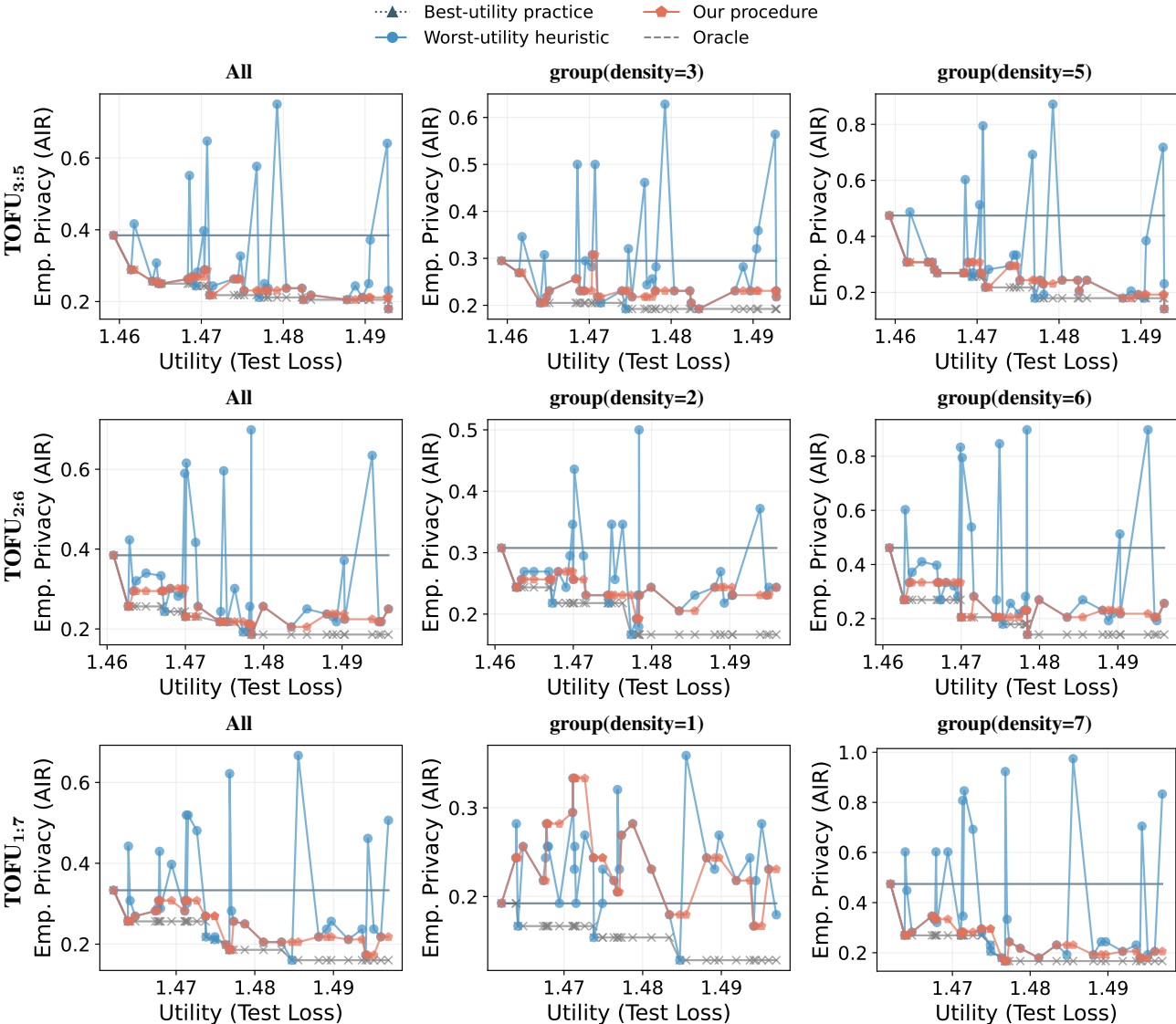

*Figure 34.* **A complete set of visualizations on the selection quality of different methods. Dataset:** *density-adjusted* **TOFU**: $\{$TOFU$_{1:7}$, TOFU$_{2:6}$, TOFU$_{3:5}\}$ (reflected in rows); Model: Llama-2-7b; $\varepsilon = 8$. Groups: all, low- and high-density groups (reflected in columns). Specifically, "density=1" means no augmentation; "density=$x$" means augmenting the dataset with additional $x - 1$ paraphrased texts besides the original one, for each sample. The advantage of **our procedure** over the others (i.e., **best-utility practice** and **worst-utility heuristic**) enlarges with the increase of the secret density. Notably, even at a small density ("density=2", meaning the secret occurs for less than 0.06% in all samples), **our procedure** already starts to demonstrate advantages.

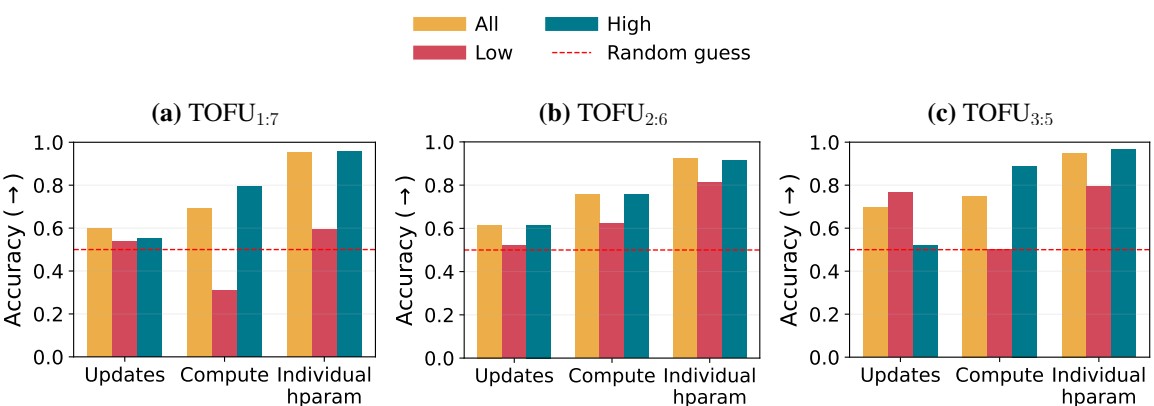

*Figure 35.* **Accuracy of the three heuristics. Dataset:** *density-adjusted* **TOFU**: $\{TOFU_{1:7}, TOFU_{2:6}, TOFU_{3:5}\}$ (reflected in columns); Model: Llama-2-7b; $\varepsilon = 8$; groups = all, low, high density groups; measure = AIR.

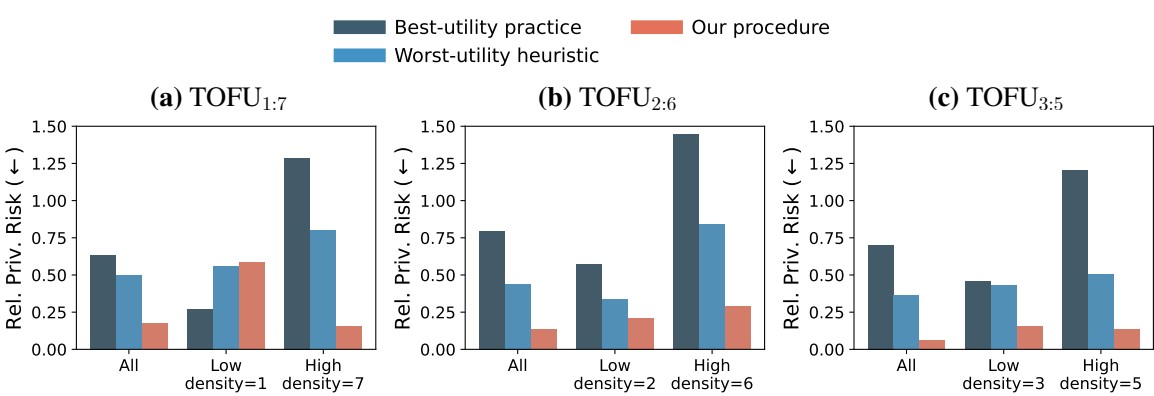

*Figure 36.* **Relative privacy risk of our procedure compared with baselines. Dataset:** *density-adjusted* **TOFU**: $\{TOFU_{1:7}, TOFU_{2:6}, TOFU_{3:5}\}$ (reflected in columns); Model: Llama-2-7b; $\varepsilon = 8$, measure = AIR, risk = rel, layout = mean. (Refer to Fig. 21 for explanations on "rel" and "mean" and Fig. 34 for explanations on "density".) Similarly, we observe that **our procedure** has a large advantage over the others, which becomes apparent even at low densities (e.g., density=2) and grows as the density increases.

