# OpenReview forum: "Empirical Privacy Variance"
_ICML.cc/2025/Conference — ICML 2025 poster_

### Official Review · Reviewer_cBBr · 2025-02-28

**Overall Recommendation:** 5

**Summary:**

For a variety of memorization based privacy attacks on language modeling, this paper studies how the average attack success can vary across hyperparameters that are meant to give the same DP guarantees. This phenomenon is observed across model sizes, architectures, datasets, and DP guarantees. Several correlations are observed, leading to proposed heuristics for choosing hyperparameters which they show improve over previous design principles at mitigating the average attack success rate. Further evidence of varying privacy guarantees is shown by instantiating privacy auditing.

## Update after Rebuttal

The authors performed experiments that showed the phenomenon was not a consequence of per-instance privacy variance, amongst answering my other concerns, and providing discussion I found very helpful. I raised my score to an accept given this.

I have further raised my score to a 5 as I believe the other reviewers concerns have been addressed. I hope this reflect my belief this paper has been conducted with significant rigour and provides valuable insights/motivation for the field.

**Claims And Evidence:**

There seems to be a subtle difference in what empirical privacy means in this paper (an average over a set), and what privacy typically means in the literature (a bound on maximum leakage). I believe being more precise will aid in future reproducibility, helps explain inconsistencies with audit attacks, and also make the motivation for policy decisions more precise. I describe this in more detail below.

Privacy is typically defined and empirically measured as a worst-case over datapoints (see auditing literature), but “empirical privacy”  in this paper is for their average (across secrets) attack success rate. Admittedly the average is over a set of secrets (or author-attribute pairs for ToFU) that was filtered and so could have implicitly been pushed to worst-case points. But a more direct method would have been to report just the worst leakage over the secrets. Moreover, at least in the case of ToFU, the filtered set still represents a quarter of the original set.

Hence, it seems it is more accurate to say this paper is studying the average of the per-instance guarantees [1] over a relatively large subset of the training datapoints, which is known to be quite different from the typical DP guarantees when using DP-SGD [2]. I would recommend the authors make the difference in their use of empirical privacy (average over a set of datapoint) to the worst-case literature clearer, as to also be clearer about the claims. This difference may also explain why their empirical privacy did not correlate with privacy audit results: the privacy audits are testing themselves per-instance guarantees over a different set of datapoints (also using a different attack). This said, as argued by the authors choice of attacks and cited literature, the behaviour of leakage for such average case users may be relevant in designing systems to not be too bad in the worst-case (DP guarantees) but also much better on average (as measured by the average in this paper).

[1] Wang, Yu-Xiang. "Per-instance differential privacy." Journal of Privacy and Confidentiality 9.1 (2019).

[2] Thudi, Anvith, et al. "Gradients look alike: Sensitivity is often overestimated in {DP-SGD}." 33rd USENIX Security Symposium (USENIX Security 24). 2024.

**Essential References Not Discussed:**

None that I noticed.

**Experimental Designs Or Analyses:**

I read Appendix E which described additional experimental details, alongside Appendix H.1 which described auditing details, and found no issues.

**Methods And Evaluation Criteria:**

The datasets used make sense for the evaluation of privacy leakage. The evaluation methods also make sense for studying average leakage over data points; privacy is typically defined as worst-case leakage over data points so the distinction with that could be made clearer (see claims section).

**Other Comments Or Suggestions:**

Below are suggestions to help clarify aspects of the paper. I look forward to hear what the authors think, and am happy to consider raising my score given their response.

1) Consider adding discussion on the differences between worst-case DP analysis, and expected privacy leakage experienced by users studied in this paper. Ideally this would be before or at the beginning of the “empirical privacy measures” paragraph to make the notion of privacy studied (in the context of the literature) more well-defined. Terms like per-instance privacy express the leakage over an individual datapoint [1], and its known these values behave quite differently to typical worst-case DP guarantees [2], posing limitations in using per-instance studies to understand worst-case guarantees (see references earlier in the review).

2) The above can also then be incorporated into the privacy standardization discussion in “why is this relevant”; I believe the argument being made in this paper is that standardization based only on worst-case leakage does not account for variation in expected leakage over datapoint, and this could now be made clearer/more precise in the text.

3) On the comparison to privacy auditing, there are two (clear) changes to the previous memorization methodology. Both methods are evaluating per-instance guarantees on different sets of points, and also using different attacks. Maybe a nice control is to run the auditing attack (or the proposed memorization attacks in the paper) on the other set of datapoints to disentangle what caused the lack of correlation (is it the attack or the set of points?). This also would shed light on the hypothesis: I believe the experiments so far have suggested the empirical privacy variations were attack independent (Figure 2), so one would hope this is still true for the loss attack.

**Other Strengths And Weaknesses:**

Strengths:
1) Experimental study is novel and thorough
2) The paper is mostly clear (see suggestion and previous comments)
3) The idea that average leakage can vary predictably with hyperparameters may have consequence to future decisions in standardizing DP

Weakness:
1) General ambiguity in what is measured in this paper by “privacy” versus DP guarantees.

**Questions For Authors:**

1) Could the authors elaborate on what the “one to many $\varepsilon$-to-risk relationship” for composition is describing? I believe for a given $\varepsilon$ and $\delta$ you can assume $\sigma$ (ideally) is chosen to be the minimal per-step noise? Unless you’re discussing changing $\sigma$ per-step, so many sequences of $\sigma$ could give the same final $\varepsilon$?

2) Could the authors also clarify what “$\varepsilon$ cannot be used for certification” means in the “why is this relevant” paragraph? In particular I found the phrasing saying a model calibrated to $\varepsilon$ cannot ensure compliance for another model confusing. I’m guessing this is a typo; does it make sense for a DP model to be used to certify another DP model?

**Relation To Broader Scientific Literature:**

Other notions of variance in privacy leakage have been observed, but these were in the context of leakage for individual data points (see a memorization study here [3]). This paper studies variation from hyperparameters for a fixed set of datapoints, and observed predictable behaviours over the hyperparameter choices. This seems novel in the context of DP to the best of my knowledge.

[3] Carlini, Nicholas, et al. "The privacy onion effect: Memorization is relative." Advances in Neural Information Processing Systems 35 (2022): 13263-13276.

**Theoretical Claims:**

No theory (and hence proofs) are presented.

---

> ### Author Rebuttal · Authors · 2025-03-29
>
> We thank the reviewer for initiating the interesting discussions.
>
> Let us begin by explaining 1) what we mean by “empirical privacy”, 2) our empirical privacy measures, and 3) the choice of computing the average score.
>
> 1) As noted in the introduction, we take a practical perspective on empirical privacy through users’ perceptions of model behaviors. This is motivated by the gap between the increasing use of DP in LLMs, and the tangible privacy risks that arise in user interactions. We believe empirical privacy should hold richer meanings in this context, extending beyond what is usually referred to (privacy attacks).
>
> 2) Our considered empirical privacy measures largely follow this standpoint. They are based on memorization and are specific to the generative nature of LLMs, rather than being generic such as the AUC of membership inference which can be applied to any ML model.
>
> 3) Finally, we average the empirical privacy scores over a set of secrets because i) this is how these metrics are evaluated in the literature (e.g., the works that proposed ACR and VMR); ii) we aim to ensure that the conclusions we draw are robust.
>
> Circling back to the reviewer’s comments,
>
> **Average- vs. worst-case.** We thank the reviewer for the insightful comment. We agree that this adds another layer of subtlety which we overlooked. We thus re-examined whether switching from average- to worst-case measures would affect our conclusions. It turns out that they remain unchanged: empirical privacy variance persists; the regression study yields similar qualitative results; the correlation between empirical privacy scores and audited values remains low.
>
> These results suggest what’s critical is not the distinction between average- and worst-case empirical privacy measures. Instead, the fundamental gap lies between **what DP promises** (preventing reidentification/membership inference) vs. **how we measure privacy** (whether a model generates a specific secret given a prompt). This gap is different from the one raised by the reviewer (per-instance guarantees vs. worst-case guarantees).
>
> Furthermore, we believe it is not appropriate to view our empirical privacy measures as representing the "average of the per-instance guarantees" or “expected leakage over datapoints” as: 1) we focus on a small subset of secrets that exhibit strong sign of memorization (Appendix E.5); and 2) a secret is only a small substring of an instance.
>
> We will add the above discussions and the new results in the camera-ready.
>
>
> **Privacy auditing.** While we agree with the reviewer that the “nice controlled study” would be great to have, there are technical challenges that prevent this. 1) Empirical privacy measures on datapoints used for auditing: Auditing uses specially crafted canary samples; they are long and each member is inserted once in the training data. Applying empirical privacy measures on them will likely yield insignificant results. 2) Auditing on extracted secrets used for empirical privacy measures: The secrets are pieces within training samples but not whole samples; they could occur in more than one sample. These create mismatches with the requirement on data in auditing. Moreover, the size of the secret set is no more than 50, while auditing typically requires a much larger set.
>
> That said, since the dataset used by the auditing method represents a worst-case dataset to maximize the lower bound the method can achieve, an alternative fair comparison is to use the worst-case empirical privacy measures, as suggested by the reviewer. Since we still observe a low correlation between empirical privacy and auditing scores, this suggests the choice of dataset is likely not an influencing factor.
>
> We believe both hypotheses we made in the “Open questions” in Sec. 5.1, 1) The auditing method is not sufficiently powerful; 2) There is a fundamental gap between membership inference and memorization-based attacks (which we also bring up in the previous point), are likely; probably even the mix of both.
>
>
> **Q1 - one-to-many relationship.** $\sigma$ is not chosen to be the minimal per-step noise but rather dependent on $b$ and $T$. It is computed using privacy accountants to satisfy a target $(\varepsilon,\delta)$-DP guarantee (Sec 2).
>
> For Laplace or Gaussian mechanism with a fixed $\delta$, the privacy budget $\varepsilon$ uniquely determines the privacy level via the single hyperparameter that controls the noise scale. In contrast, the compositional nature of DP-SGD allows infinitely many configurations to achieve the same $\varepsilon$, each yielding its own privacy level. We intend to highlight that empirical privacy variance is a unique characteristic of DP-SGD, and more broadly, of DP algorithms that involve composition.
>
> **Q2 - certification.** As illustrated in Fig 4, all models in the red region (with $\varepsilon < \varepsilon^\star$) will fail to pass the privacy tests. Please refer to our last response to Reviewer u6Yf for more details.

---

> > ### Comment · Reviewer_cBBr · 2025-04-02
> >
> > Thanks for the response!
> >
> > I'm happy with the clarification on what was causing the mismatch (if it was a per-instance observation). I believe the paper can be quite strong with this added discussion, which provides more evidence on the claim (a mismatch between what we are measuring and what privacy is promising). I look forward to the additional figures and discussion in the revised paper, and have raised my score.
> >
> > Just a clarification, could the authors provide some details on what the worst-case measures they tested were? I'm guessing it's the datapoint/secret with the most leakage?

---

> > > ### Author Response · Authors · 2025-04-02
> > >
> > > We thank the reviewer for acknowledging our response and raising the score. We will definitely include these discussions & new results in our revision, and also reflect in our acknowledgement.
> > >
> > > To reply to the reviewer’s question – yes, the worst-case measure evaluates the highest leakage over the set of secrets, basically, switching the aggregation calculation from the mean over the secrets (used in the original average-case measure) to the maximum over the secrets. As an additional note, the set of secrets we curated already represent those with high leakage.

---

### Official Review · Reviewer_MX8d · 2025-03-03

**Overall Recommendation:** 4

**Summary:**

This paper uses the concept of empirical privacy variance to show that models trained with DP-SGD under the same $(\epsilon, \delta)$ guarantee but using different hyperparameter settings can yield varying empirical privacy. Empirical variance metrics are defined that quantify how much a model memorizes information. The experiments reveal varying scores under these metrics depending on different hyperparameter configurations, with generality across multiple dimensions of the problem (different empirical variance metrics, secret subsets, model sizes...). Regression analysis is used to measure the relationship between DP-SGD hyperparameters (including joint effects) and the empirical variance score. Practical hyperparameter selection heuristics are proposed to mitigate this phenomenon. Finally, hypotheses on the cause of empirical privacy variance are discussed.

**Claims And Evidence:**

The claims in the paper are supported by well-structured and clear evidence. I believe the experimental evidence in the main paper is sufficiently articulate and convincing. I did not review supplementary results in the Appendices in detail.

**Essential References Not Discussed:**

I'm not aware of essential references that aren't cited. However, I think the short section comparing the paper to related work in Appendix A should be discussed in the main paper, with a dedicated and more articulated section. That would make it easier for the reader to locate this paper in the literature landscape. I am aware that the authors chose to present that in Appendix at this stage due to space constraints, and I encourage them to include a more extended presentation of the related work in future iterations.

**Experimental Designs Or Analyses:**

I checked the soundness and validity of all experiments presented in the main paper, which I find to be well-structured, relevant to the proposed question, and clearly presented.

**Methods And Evaluation Criteria:**

The chosen evaluation metrics, models and data sets are appropriate for answering the research question explored in the paper.

**Other Comments Or Suggestions:**

I don't have additional comments.

**Other Strengths And Weaknesses:**

The main strength of the paper is that it highlights an interesting and general no-free-lunch problem arising from hyperparameter setting selection in DP-SGD. To the best of my knowledge, this is the first time empirical privacy variance is proposed and investigated on a practical level. The experiment results provide compelling evidence that the problem is general, and likely to impact a variety of models using DP-SGD. For this reason, I do think that the implications of empirical privacy variance are potentially broad and of interest to the DP community. This paper poses the basis for interesting future exploration, while also providing practical heuristics to mitigate the problem.

One limitation of the paper is that empirical privacy might need to be measured differently depending on the specific task at hand, and it is unclear to which extent the proposed heuristics apply and generalize. This might be better clarified by including a limitations section in the next iteration of the paper.

**Questions For Authors:**

I encourage the authors to address the minor points I brought up in the review, particularly the inclusion of a Limitations section and a more detailed Related Work section in the paper.

**Relation To Broader Scientific Literature:**

The paper proposed tools to examine the phenomenon of empirical privacy variance in DP-SGD, and analyses the case of private fine-tuning of language models. The paper is located in the landscape of DP-SGD literature. DP-SGD was first introduced by Abadi et al. (2016). The impact of hyperparameter configuration on model utility has been explored among others by Ponomareva et al. (2023). This paper focuses on the effects of hyperparameter choice on empirical privacy instead. The works most closely related to this paper are mentioned in Appendix A: Hayes et al. (2023) show how different hyperparameters impact the success rate of reconstruction attacks; Kaissis et al. (2024) propose a metric to quantify the maximum excess vulnerability among mechanisms that share the same privacy budget. These works are focused on theoretical analysis of such problems, while this paper is focused on a practical, real-world setting (privately fine-tuned language models) and proposes heuristics and intuitive hypothesis on potential causes of this phenomenon.

**Theoretical Claims:**

There are no theoretical proofs in the paper. The main claims are supported by empirical evidence.

---

> ### Author Rebuttal · Authors · 2025-03-29
>
> We thank the reviewer for recognizing our claims and methods as clear, sound and relevant, and our contributions to be of broad interest to the community. We will take the reviewer’s feedback and incorporate a more detailed related work section into the main body, incorporating discussions in our response to Reviewer E9Wi. We will also include a limitation section discussing what has not been covered in our work but deserves attention: in particular, the subtle difference between shuffling and Poissong subsampling (which is currently mentioned in a footnote), as well as the privacy cost of hyperparameter tuning.
>
> Regarding the reviewer’s comment on the generality of heuristics, we clarify that while they are developed based on regression analysis in two specific scenarios (fine-tuning GPT-2-S on Enron at $\varepsilon=4$ with the target variable as ACR, and Llama-2-7b on TOFU at $\varepsilon=16$ with the target variable as AIR), our experiments demonstrate that they generalize across various models, datasets, privacy budgets, and empirical privacy measures (VMR). The detailed results are presented in Figs. 7 and 10 in the main paper, as well as Figs. 18, 20, 21, and 22 in Appendix G. In total, we have spent over 10,000 H100 GPU hours to ensure that our findings are general and robust. We hope this convinces the reviewer about the broad applicability of these heuristics. Please let us know if you have further questions or suggestions. Thank you!

---

### Official Review · Reviewer_u6Yf · 2025-03-10

**Overall Recommendation:** 2

**Summary:**

The paper empirically estimates the privacy loss of language models fine-tuned with DP-SGD in many different configurations, including different hyperparameters, model sizes, and dataset characteristics. The paper finds that models calibrated to the same DP guarantee can have very different empirical privacy losses depending on the full configuration, and calls this phenomenon "empirical privacy variance". The paper investigates the effect of hyperparameters further, and finds rules of thumb on how the hyper parameters change empirical privacy loss that are derived from experimental results, and generalise in additional experiments. The paper also investigates possible causes of empirical privacy variance, but does not find a clear cause.

## update after rebuttal

I'm keeping my score due to the subsampling discrepancy in the paper. I have read the extensive final response from the authors. I agree with the authors that there are reasons to believe the conclusions regarding the subsampling rate in Section 4 would not change with correct privacy accounting, but if these were to be included in the paper, they would need a massive caveat that basically says "we don't really know if these conclusions are correct". I also agree that training any kind of large model with Poisson subsampled DP-SGD is very difficult in practice, which is why I suggested recalculating the privacy bounds for the models that have already been trained.

However, I disagree with the other arguments from the authors, which boil down to arguing that the paper should be accepted because previous papers with the same issue have also been accepted. This is not a good argument, as consistently applying it would mean that bad research practices cannot be eliminated if they are ever accepted.

In summary, I think the paper could be accepted if the results are presented properly, which is why my score is a 2 instead of a 1. However, this presentation would be very different from the current one, and include many limitations and caveats not in the submission, so I'm keeping my score at a 2 instead of a 3.

**Claims And Evidence:**

The paper's claims are supported by extensive experiments. However, the mismatch between shuffling-based subsampling and Poisson subsampling is a major issue with the experiment design, more details in the specific section.

**Essential References Not Discussed:**

No essential references missing.

**Experimental Designs Or Analyses:**

The paper does not discuss the implications of the mismatch between using shuffling-based subsampling in practice and Poisson subsampling in privacy accounting that the paper's experiments suffer from. As noted by Chua et al. (2024b;c), there is a difference between the subsampling amplification that the two subsampling schemes provide, leading to different privacy bounds if accounted for correctly. As a result, all $\epsilon$ values shown in the paper are incorrect.

This does not substantially change many results of the paper. For example, the $\epsilon$ values in Figure 2 would be adjusted, but all other points would remain the same, and the conclusions drawn from the figure would not change. However, the results examining the effect of the batch size could be affected, since this effect could depend on the subsampling scheme.

As is, I can't recommend accepting a paper where all $\epsilon$ values are incorrect, and the only mention of this issue is a footnote that brushes this off as something outside the scope of the paper. However, given the otherwise high quality of the paper, I could recommend accepting if this issue were prominently discussed, for example by making it clear that the $\epsilon$ values are only estimates, and that the batch size may not behave the same way with a correctly computed $\epsilon$. Alternatively, you could do the correct privacy accounting manually, since doing that with parallel composition is fairly simple. See the start of Section 3 in Chua et al. (2024c) for the analysis with deterministic subsamples, and Theorem 4.1 in Chua et al. (2024b) for a proof that the same analysis gives a valid privacy bound for any shuffling.

**Methods And Evaluation Criteria:**

The ACR metric seems to potentially underestimate the privacy loss for the secrets in the Enron dataset, since the metric only looks (as I understand it) for reproductions of the exact secret string, not other strings that would reveal the same secret. For example, it may be easier to get a model to produce the string "Richard B Sanders/HOU/ECT" than the full secret "Forwarded by Richard B Sanders/HOU/ECT". In this case, ACR would be based on the prompt that produced the latter string, though the former reveals almost as much.

**Other Comments Or Suggestions:**

- In Figure 3, the "increasing dataset size" is easy to interpret as the x-axis label for the top row.
- Lines 267-270 (left): the sentence "a model calibrated to a given $\epsilon^*$, deemed to meet privacy requirements" is confusing. I initially interpreted this to mean that $\epsilon^*$ is deemed to meet privacy guarantees, but the argument only makes sense if the model is deemed to meet privacy guarantees independently of the $\epsilon^*$.
- In Figure 5, the x-axis label is confusing, since higher utility should be better, but higher test loss is worse.
- Figure 9 is too small to read easily.

**Other Strengths And Weaknesses:**

The topic of the paper is important. The results follow a general pattern that has been found in existing work, but the empirical privacy perspective on this pattern is both novel and important.

**Questions For Authors:**

No questions.

**Relation To Broader Scientific Literature:**

The relevant literature is discussed for all findings of the paper.

**Theoretical Claims:**

The paper does not make theoretical claims.

---

> ### Author Rebuttal · Authors · 2025-03-29
>
> We thank the reviewer for raising the subtlety between implementing DP-SGD with shuffled batches, but performing privacy accounting as if Poisson subsampling was used.
>
> In the camera-ready version, we are happy to expand the discussion and move it to the main paper under a “limitation” section. Additionally, we would like to clarify a few more points.
>
> While as pointed out the exact $\\varepsilon$ values are incorrect, we believe that the qualitative observations we make are still relevant. For example, in the plots where we have $\\varepsilon$ on the x-axis e.g. Figures 2, 3 and 10, if the correct privacy accounting of shuffle-based DP-SGD were to be applied for the specific parameter settings we have, **it would only re-scale the x-axis in some (non-linear) manner**, but the empirical variance of the y-axes will remain the same which is precisely the main take-away from this paper. That is, the x-axis, instead of being supported on $\\varepsilon$ in $\\{1, 2, 4, 8, 16\\}$, would be supported on some other different values, though still monotonically increasing.
>
> Thus, while the observations made by Chua et al. (2024b;c) are valuable, we consider them **relevant primarily if some model is released after training on truly sensitive data, with a claimed privacy guarantee that is incorrect.** But in our case, we feel this observation is not that critical, as it just amounts to some rescaling of values which do not affect the qualitative conclusions from our study. But thank you for raising this; we will elaborate more on this in the revision.
>
> Additionally, we note that Chua et al. (2024c) was posted on arXiv on Nov 6, 2024, which is within three months of the ICML 2025 submission deadline (Jan 30, 2025). According to [the ICML 2025 reviewer instructions](https://icml.cc/Conferences/2025/ReviewerInstructions),  this qualifies as a **“concurrent work”** that “Authors are not expected to discuss”. In fact, while we are aware of the earlier work Chua et al. (2024b), which highlights the gap between these two sampling approaches, we were not aware of any efficient implementations of Poisson subsampling for DP fine-tuning LLMs: 1) Chua et al. (2024c) was not available online when we initiated our work; 2) They conducted experiments on MLPs only, and extending the implementation to transformer training is non-trivial and warrants further research. As a consequence, we adhere to the conventional DP fine-tuning framework for LLMs that employs shuffle-based DP-SGD.
>
> Since this was the main concern raised in the review, we hope the reviewer could reconsider the rating if the above response adequately addresses their concern. We are happy to answer any other questions during the discussion period if it helps to clarify this.
>
> > For other minor comments--
>
> **ACR underestimates the privacy loss.** In our studies, the measured empirical privacy scores depend heavily on the chosen set of secrets. The reviewer is correct that using a shorter (sub)string as the secret or not requiring an exact match could result in a lower measured privacy loss. However, the magnitude of these scores is not our primary focus; rather, our arguments center on the **variance** of the empirical privacy scores.
>
> **Figures.** Thanks for the suggestions, we will incorporate them in the camera-ready version.
>
> **Clarifications on “deemed to meet privacy requirements”.** Here's what we intend to convey: even if a legislative body determines that a model with a privacy budget $\varepsilon^\star$ passes their privacy tests, this does not imply that models with a stricter privacy budget ($\varepsilon \le \varepsilon^\star$) will pass. For instance, as illustrated in Fig 4, all models in the red region will fail to pass the privacy tests.

---

> > ### Comment · Reviewer_u6Yf · 2025-04-02
> >
> > Thank you for the response.
> >
> > **Subsampling discrepancy**
> >
> > I agree that many of your conclusions would not change if you did privacy accounting correctly, and adding discussion of this to a limitations section is a good way to make this clear to readers. However, the conclusions concerning the batch size could change, so I don't think they should be included in the paper without correct privacy accounting. This would effect a lot of the results in Section 4, so I'm hesitant to recommend accepting the paper without any reviewers seeing such major changes.
> >
> > **Clarifications on “deemed to meet privacy requirements”**
> >
> > I think your new version of explaining this still needs clarification. I interpreted this as:
> > - Legislative body has chosen empirical privacy tests that do not look at $\epsilon$ directly.
> > - Model A that is $\epsilon^*$-DP passes privacy tests.
> > - Model B with $\epsilon \leq \epsilon^*$ can still fail privacy tests.
> >
> > If this is correct, I think you need to make it clear that the privacy tests do not look at $\epsilon$ directly. Even from your newer version, one can easily get the impression that the privacy test is just "is $\epsilon$ small enough".

---

> > > ### Author Response · Authors · 2025-04-02
> > >
> > > Thanks for the suggestion on the second point; we will expand the discussion. In the paper, we interpret “privacy tests” as evaluations of empirical privacy, as this is the main focus of our study.
> > >
> > > Regarding the subtlety between shuffling and Poisson subsampling and its potential impact on the effect of sampling rate $q$, our intuition is that the conclusion is unlikely to change with a different sampling scheme:
> > >
> > > 1. Hayes et al. (2023) theoretically show that the success rate of a tight upper of reconstruction attacks increases with the sampling rate, reconciling with our findings.
> > >
> > > 2. Prior work shows that larger batch sizes improve utility. It is more plausible that this improvement comes at the cost of empirical privacy.
> > >
> > > 3. Chua et al. (2024c) suggest that utility of models trained under DP-SGD with Shuffle and Poisson are basically the same, and we expect this to hold for empirical privacy.
> > >
> > > Nevertheless, we acknowledge that we cannot **entirely** rule out the possibility that switching from shuffling to Poisson subsampling might alter the relationship between empirical privacy and $q$. That said, we wish to emphasize the following:
> > >
> > > 1. “Reporting DP guarantees under Poisson subsampling while training with shuffled batches” is a **common** (albeit inaccurate) practice in the community. This dates back to Abadi et al. (2016), and has since been adopted in most papers on large-scale DP training. Below is an incomplete list of references confirmed to use shuffling.
> > >
> > > - Yu et al. "Differentially Private Fine-tuning of Language Models." ICLR 2022. [dp-transformers](https://github.com/microsoft/dp-transformers)
> > >
> > > - Li et al. "Large Language Models Can Be Strong Differentially Private Learners." ICLR 2022. [code](https://github.com/lxuechen/private-transformers) with Opacus==0.13.0 which implemented shuffling-based sampler
> > >
> > > - Yue et al. "Synthetic Text Generation with Differential Privacy: A Simple and Practical Recipe." ACL 2023. Implementation follows dp-transformers.
> > >
> > > - Yu et al. "Privacy-Preserving Instructions for Aligning Large Language Models." ICML 2024. [code](https://github.com/google-research/google-research/tree/master/dp_instructions)
> > >
> > > - Zhang et al. "DPZero: Private Fine-Tuning of Language Models without Backpropagation." ICML 2024. [code](https://github.com/Liang137/DPZero)
> > >
> > > - Panda et al. "Privacy auditing of large language models." ICLR 2025. Implementation based on [code](https://github.com/awslabs/fast-differential-privacy) which uses shuffling-based sampler
> > >
> > > - McKenna et al. "Scaling Laws for Differentially Private Language Models." arXiv:2501.18914. [paper](https://arxiv.org/pdf/2501.18914) Appendix A, paragraph “Fixed Physical Batch Size”.
> > >
> > > 2. We are not aware of efficient implementations of Poisson subsampling for transformers. The need to accommodate variable batch sizes forces memory allocation based on worst-case scenarios, significantly reducing efficiency and complicating load balancing. It further exacerbates the memory strain for DP fine-tuning in LLMs. Opacus lacks built-in support for advanced memory management and distributed training strategies that transformers demand, making them unsuitable for direct use on LLMs without substantial modifications.
> > >
> > > 3. The reviewer suggested the privacy accountant for deterministic batch sampling in Chua et al. (2024c). While this provides an upper bound on the privacy loss, it is conservative as it neglects the amplification effect from shuffling. This is why people rarely rely on such accountants [link](https://arxiv.org/abs/2208.04591) in practice. We believe this is more appropriate in scenarios where the goal is to **release** a model with a formal privacy guarantee. However, our objective is to **calibrate** models to the same target guarantee for comparative analysis. Using a loose bound would undermine this calibration and distort the interpretation of our empirical findings.
> > >
> > > We must point out that if the reviewer applies the standard uniformly, all the cited works could be rejected on the grounds “all ε values are incorrect.” This would be an unreasonable stance. Also, the reviewer's critique would discredit major conclusions from prior work, e.g., Sec 3 of Li et al. (2022) study hyperparameter tuning and make recommendations to use large batch size. Disregarding findings due to nuances in sampling schemes while ignoring the broader insights and methodological value would make no progress possible.
> > >
> > > We recognize that DP-ML is a broad field and contains nuances that need to be seriously taken and addressed. The distinction between sampling schemes is important, and we appreciate the progress in Chua et al. (2024b;c). However, the technical challenges involved (e.g., building a framework for LLM) are not something that can be resolved overnight.
> > >
> > > We respectfully ask the reviewer to adopt a consistent and constructive standard—one that acknowledges the context, intent, and scientific value of each contribution.

---

### Official Review · Reviewer_E9Wi · 2025-03-14

**Overall Recommendation:** 1

**Summary:**

The paper investigates privacy implications when fine-tuning language models using DP-SGD. Importantly, when using DP-SGD, the same (ε, δ)-DP guarantee can be achieved through multiple hyperparameter configurations (batch size, number of iterations, learning rate, etc.).
The authors' key finding is that despite having identical theoretical privacy guarantees, models trained with different hyperparameter configurations exhibit variations in empirical privacy (which they measure through memorization metrics). This "empirical privacy variance" means that the theoretical (ε, δ) parameters alone don't fully characterize the practical privacy leakage of the model, raising questions about the practical interpretation of DP guarantees in language model fine-tuning. Authors suggest some new heuristics to hyper-parameter tune the parameters of DP-SGD so that we get a better trade-off between utility and empirical privacy.

**Claims And Evidence:**

- The main claim of the paper is that there is a variance in "empirical privacy" when the "theoretical privacy" is fixed. This claim is supported by their extensive experiments.

- There are other claims about the role of dataset size and model size and their role in empirical privacy. These claims motivate their hyperparameter selection. I think they have moderately supported these claims but for suggesting general heuristics for hyperparameter turning I would expect more empirical validation.

- One of the claims of the paper which I did not find a support for is that the choice of hyperparameters is the main factor explaining the variance. However, I think there could be other sources contributing to this variance. For example the attacks might be more successful for smaller models. Or there could be variance stemming from the choice of dataset.

**Essential References Not Discussed:**

-

**Experimental Designs Or Analyses:**

Their experimental design makes sense to me.

**Methods And Evaluation Criteria:**

The method for training private models (DP-SGD) and also their empirical privacy evaluation methods (ACR and VMR) make sense.

**Other Comments Or Suggestions:**

See above

**Other Strengths And Weaknesses:**

- The first weakness with the paper is that there is an implicit assumption that the source of variance is the selection of hyperparameters. I think the methods used for privacy evaluation are actually more of a source for variance. For example, ACR itself has a very brittle optimization step that could fail in finding the optimal solution (it does most of the time). I


- argue that ACR and VMR should not be set as a goal. It is only a metric that could indicate privacy violations. Having a low ACR is not at all and indication of privacy. So I argue performing hyperparameter tuning to lower these rates does not necessarily lead to more private models.

- The main contribution of paper seems to be about showing the existing variance. As authors argue, this is already discussed in the literature.

- Writing of the paper is confusing. I do not understand many sentences in the paper.

- As a strength, I find the empirical studies in the paper rigorous and impressive.

**Questions For Authors:**

- I did not understand this sentence: "A direct consequence is that, in DP-SGD, ε cannot be used for certification: a model calibrated to a given ε*, deemed to meet privacy requirements, cannot ensure compliance formodels with stricter DP guarantees (ε ≤ ε∗)." Could you please elaborate?

- Is there a theory that suggests ACR and VMR should be small for differentially private mechanisms? Your seem to be assuming this but I have not seen a theoretical analysis on this.

- The two hypothesis in section 5.1 are not clear. Can you please elaborate? The first hypothesis is somewhat clear, you are saying that there is gap in our understanding of true epsilon values. But I have a hard time understanding the second hypothesis.

- I don't understand this sentence, please elaborate: Why is this relevant? Consider classic DP mechanisms such as the Laplace and Gaussian mechanisms (Dwork et al., 2014). Their noise parameter (scale parameter b for Laplace and σ for Gaussian) inversely correlates with ε and uniquely determines privacy risk: increasing it lowers the signal-tonoise ratio, making it harder for adversaries to extract meaningful information. This establishes a one-to-one, monotonic ε-to-risk relationship. In contrast, the composition nature of DP-SGD results in a one-to-many ε-to-risk relationship, making ε insufficient to fully capture privacy risk.

**Relation To Broader Scientific Literature:**

As stated by authors, the finding that same privacy parameters could lead to different implications for empirical attacks is already studied in the literature. I think authors need to clarify their contribution in comparison with these works a bit better.

**Theoretical Claims:**

- N/A

---

> ### Author Rebuttal · Authors · 2025-03-29
>
> Thanks for the review.
>
> > Summay of our key points
>
> - We measure empirical privacy in controlled settings
> - We conduct extensive experiments (> 10,000 H100 GPU hours) for robust findings
> - Our contributions go well beyond “showing the existing variance”
> - We recommend empirical privacy as an additional dimension of privacy alongside DP, rather than targeting it in isolation
>
> > Detailed responses
>
> **Sources of variance.** The empirical privacy variance we observe is **conditioned on** fixing the dataset, model, training algorithm and empirical privacy measure (Sec 3.3 para 1). We also show variance from *training randomness* is small (Appendix F.3).
>
> The reviewer raised variance in *measuring empirical privacy scores*. We believe this is not the case for our experiments. For ACR, we run the optimization with 3 seeds and report the highest score for more accurate estimates; VMR and AIR average 10 stochastic decodings to reduce variance (Appendix E.6).
>
> **Comparison with related works.** Appendix A discusses related work.
>  Hayes et al. (2023) and Kaissis et al. (2024) show the attack success rate of a reconstruction attack (which relies on strong assumptions) can vary for mechanisms calibrated to the same DP guarantee. We focus on realistic privacy risks that emerge in language model interactions, leading to the concept of empirical privacy variance.
>
> Beyond this, we have more substantial contributions: 1) extensive experiments across various models, datasets, secrets, and empirical privacy measures; 2) in-depth qualitative analysis of the impact of hyperparameters and investigation of the cause; and 3) broader implications: for practitioners, we provide heuristics that enhance empirical privacy; for researchers, we expose the hidden cost of hyperparameter tuning; for policy makers, we discuss how this phenomenon could complicate standardization.
>
> As Reviewer MX8d noted, “...this is the first time empirical privacy variance is proposed and investigated on a practical level. The experiment results provide compelling evidence that the problem is general, and likely to impact a variety of models using DP-SGD… the implications of empirical privacy variance are potentially broad and of interest to the DP community.”
>
> Given the above, we kindly ask the reviewer to reconsider the rating based on our contributions.
>
> **More empirical validation.** Figs. 7 and 10 and Appendix G (Figs. 18, 20-22) confirm our heuristics’ generality across models, datasets, $\varepsilon$’s and empirical privacy measures. Could the reviewer kindly suggest additional evidence they would like to see?
>
> **Issues with ACR/VMR.**  While ACR has limitations, we believe a lower ACR/VMR score indicates lower memorization and thus better privacy **in general**. Importantly, we don’t advocate for targeting these **in isolation**. Our heuristics operate on configurations calibrated to a specified DP guarantee (Sec 4.3), adding a dimension not captured by DP.
>
> **Q1 - certification.** We mean: even a legislative body determines that a model with $\varepsilon^\star$ passes their privacy tests, this does not imply that models with a stricter privacy budget ($\varepsilon \le \varepsilon^\star$) will pass. As illustrated in Fig 4, all models in the red region will fail.
>
> **Q2 - theory for ACR/VMR.** The statement “ACR and VMR are small for DP mechanisms” is incomplete: 1) “small” is not quantitatively defined; 2) the privacy parameters of the DP mechanism are not specified. We do not have a theory for it—as we point out in the caption of Fig. 4, bridging DP and empirical privacy measures is generally hard.
>
> The reviewer might be suggesting that without theoretical guarantees, a model with DP can have large ACR/VMR, so our findings are not surprising. But we want to emphasize: what we believe makes our finding interesting is the **simultaneous existence** of both high and low ACR/VMR under the same theoretical privacy budget (see Fig. 2). This naturally motivates our study in Sec. 4 to target the low end by proper hyperparameter selection.
>
> **Q3 - hypothesis in Sec 5.1.** The hypothesis here is two-fold. First, while all mechanisms are calibrated to the same $(\varepsilon, \delta)$-DP, the final LLMs (where we measure empirical privacy) might have different “real” $\varepsilon$’s, all upper bounded by $\varepsilon$. Second, these “real” $\varepsilon$’s reflect empirical privacy: a higher value corresponds to poorer empirical privacy.
>
> **Q4 - why relevant.** For Laplace or Gaussian mechanism, $\varepsilon$ uniquely determines the privacy level via the single hyperparameter that controls the noise scale. In contrast, the compositional nature of DP-SGD allows infinite configurations to achieve the same $\varepsilon$, each with unique privacy levels. Fig. 4 highlights that empirical privacy variance is a unique characteristic of DP-SGD, and more broadly, of DP algorithms that involve composition.
>
> Please let us know if you have further questions. Thank you!

---

### Decision · Program_Chairs · 2025-05-01

**Decision:**

Accept (poster)

**Comment:**

The paper presents comprehensive experiments exploring empirical privacy measures, such as secret memorization, in LLMs trained with different hyper-parameters to give the same formal privacy guarantees. They find that there is significant variation in empirical privacy across these models, identify trends in how the hyper-parameters relate to empirical privacy, and discuss possible causes.

All reviewers agreed that the experiments were rigorous and impressive, with various smaller questions about the findings or interpretation, most of which were well answered by the rebuttal.

Reviewer E9Wi raised concerns about novelty with respect to the observation that practical privacy according to some attack model could vary over models satisfying the same (epsilon, delta)-DP guarantee. It appears the most important precedent Hayes et al. 2023. However, the submission expands significantly by performing an expansive set of experiments with robust findings on a specific, important, application area (LLMs) with tailored privacy metrics based on memorization (and, for example, not assuming an informed adversary that knows all records except for one, as in Hayes et al.), and by identifying specific trends in the results. This reinforces the prior work and extends its findings to provide a broad empirical foundation for empirical vs formal privacy in LLMs that is likely to inform future research.

Reviewer u6Yf's primary concern is the discrepancy between analysis based on Poisson subsampling but an implementation based on shuffling. Apart from this point, the reviewer thinks the paper is "otherwise high quality" and would recommend acceptance. I appreciate the reviewer bringing up this point, since recent evidence (Chua et al., 2024) indicates it is important to address. However, the practice has been a norm in the field for almost a decade, so it will take some time to make the change. Further, both the reviewer and authors agreed it would have a very minor impact on most of the findings. I encourage the authors to address this more prominently in writing in the final version of the paper.

Reviewer MX8d was largely satisfied with the work, but recommends a more prominent discussion of related work. I (and the authors, in their rebuttal) agree with this point.

Reviewer cBBr brought up a specific issue of worst-case vs. average privacy over instances. The authors ran additional experiments to address this issue, and the reviewer raised their score. They raised their score further because they believed the authors had addressed the concerns raised by other reviewers.

Overall, the paper appears to be a solid empirical contribution that can inform the discrepancy between theoretical and practical privacy in LLMs.